

# Mixed layer depth variability in the Red Sea

Cheriyeri P. Abdulla[1][*], Mohammed A. Alsaafani[1, 2], Turki M. Alraddadi[1], and Alaa M. Albarakati[1]

[1]Department of Marine Physics, Faculty of Marine Sciences, King Abdulaziz University, Jeddah, Saudi Arabia.
[2]Department of Earth & Environmental Sciences, Faculty of Science, Sana'a University, Yemen.

*Correspondence to*: Cheriyeri P. Abdulla (acp@stu.kau.edu.sa)





## Abstract

For the first time, a monthly climatology of mixed layer depth (MLD) in the Red Sea has been derived based on temperature profiles. The general pattern of MLD variability is clearly visible in the Red Sea,

with deep MLDs during winter and shallow MLDs during summer. Transitional MLDs have been found during the spring and fall. Northern end of the Red Sea experienced deeper mixing and higher MLD, associated with the winter cooling of the high-saline surface waters. Further, the region north of 19° N experienced deep mixed layers, irrespective of the season. Wind stress plays a major role in the MLD variability of the southern Red Sea, while net heat flux and evaporation are the dominating factors in the

central and northern Red Sea regions. Ocean eddies and Tokar gap winds significantly alters the MLD structure in the Red Sea. The dynamics associated with the Tokar gap winds leads to a difference of more than 20 m in the average MLD between the north and south of the Tokar axis.

**Keywords:** Mixed layer depth, Red Sea, Eddies, Tokar gap winds, Air-Sea interaction.



# 1 Introduction

Surface mixed layer is a striking and universal feature of the open ocean where the turbulence associated with various physical processes leads to the formation of a quasi-homogeneous layer with nearly uniform properties. The thickness of this layer, often named mixed layer depth (MLD), is one of the most important oceanographic parameters, as this layer directly communicates and exchanges energy with the atmosphere and therefore has a strong impact on the distribution of heat (Chen et al., 1994), ocean biology

(Polovina et al., 1995) and near-surface acoustic propagation (Sutton et al., 2014). Heat and fresh-water exchanges at the air-sea interface and wind stress are the primary forces behind turbulent mixing. Similarly, stirring associated with turbulent eddies predominantly changes the mixing process, mainly along the isopycnal surfaces where stirring may occur with minimum energy (de Boyer Montégut et al., 2004; Hausmann et al., 2017; Kara et al., 2003).

The oceanic heat loss cools the mixed layer and weakens the stratification, leading to strong mixing and a deeper MLD. Similarly, the heat gain warms the mixed layer and strengthens the stratification, leading to weak mixing and shallow MLDs. The fresh-water loss makes the surface water more saline and denser, leading to enhanced mixing and deeper MLDs, while the fresh-water gain makes the surface water fresher and lighter, leading to diminished mixing and shallow MLDs. The momentum transmitted to the ocean

through from the wind stress acts as the primary dynamic force for the upper layer turbulence and circulation. The shear and stirring generated by the wind stress enhance the vertical mixing and play a major role in controlling the deepening of the oceanic mixed layer. In some regions, the mixed layer variability is mainly controlled by wind stress.

The Red Sea is a typical semi-enclosed narrow basin connected to the Indian Ocean through the Gulf of

Aden. It is one of the important deep water formation regions, and its signature reaches into the Indian Ocean (Beal et al., 2000). The Red Sea is surrounded by extremely hot arid lands and has a relatively strong evaporation rate (2 m yr$^{-1}$) with nearly zero precipitation (Albarakati and Ahmad, 2013; Sofianos et al., 2002). This region experiences strong seasonality in its atmospheric forcing and buoyancy. These characteristics, along with the lack of river input, make the Red Sea one of the hottest and most saline

regions in the world. The narrow and semi-enclosed nature of the basin, the presence of multiple eddies, strong evaporation, lack of river input and very weak precipitation, seasonally reversing winds, etc. lead to complex dynamical processes in the Red Sea (Aboobacker et al., 2016; Zhai and Bower, 2013; Zhan et al., 2014).

The increase in temperature and salinity profiles in recent years enhanced the study of MLD structure and

its variability, both globally (de Boyer Montégut et al., 2004; Kara et al., 2003; Lorbacher et al., 2006) and regionally (Abdulla et al., 2016; D'Ortenzio et al., 2005; Keerthi et al., 2012, 2016; Zeng and Wang, 2017). The Red Sea has been investigated for many years with an emphasis on its different physical features. But, no detailed investigation on MLD variability has been documented so far in the Red Sea, except few studies addressing the hydrography and vertical mixing of localized areas, such as the Gulf of

Aqaba (Carlson et al., 2014) and the Bab-el-Mandab region (Alsaafani and Shenoi, 2004).

In this work, an MLD climatology is produced for the first time based on in situ observations. Further, the roles of atmospheric forces and oceanic eddies on the changes of the MLD have been investigated. The following sections are arranged as: Sect. 2 describes the datasets used and methodology. The subsequent sections discuss the observed MLD variability in the Red Sea (Sect. 3), the role of the major

forces on the MLD variability (Sect. 4), the impact of eddies on MLD changes (Sect. 5) and the influence of Tokar gap winds (Sect. 6). The main conclusions of the present work are given in the final section.

## 2 Data and methods

### 2.1 Datasets

Temperature and salinity profiles from different sources are collected, which are measured using CTD

(conductivity-temperature-density profiler), PFL (autonomous profiling floats including ARGO floats), XBT (expendable-bathy-thermograph) and MBT (mechanical-bathy-thermograph). The World Ocean Database (https://www.nodc.noaa.gov/OC5/SELECT/dbsearch/dbsearch.html) is the main source with larger number of profiles. Apart from this, data from Coriolis data center





(http://www.coriolis.eu.org/Data-Products/Data-Delivery/Data-selection) and several cruises conducted
by individual institutions are also used in this analysis. The bathythermograph profiles were depth-
corrected based on Cheng et al., (2014). A total 13,891 temperature profiles were made for the Red Sea
(approximately 14 % of these profiles have salinity measurements) from 1934 to 2017.

These profiles are quality checked according to the procedure given in Boyer and Levitus (1994). In the
duplicate check, all the profiles within a 1 km radius and taken on the same day are considered duplicates
and are removed from the main dataset. The levels in the profile with large inversions in temperature
(inversion $>= 0.3°$ C) are flagged and removed. If three or more inversions are present, then the entire
profile is removed. The levels with extreme gradients $>=0.7°$ C are also removed from the profile. The
number of profiles removed at each step of the quality check is tabulated in Table S1. Since the present
work is more focused on the changes in the upper layer of the ocean (from the surface to a 150 m depth),
profiles with low resolutions in the upper layers are removed. Almost 50 % of the profiles have resolutions
of <5 m, while 7 % of the profiles have poor resolutions (resolutions of > 25 m).

A total of 11,212 profiles passed the quality check from CTD (690), PFL (1385), XBT (5507) and MBT
(3630), and the spread is shown in Fig. 1. More than 80 % of these profiles are positioned along the middle
of the Red Sea, with a sufficient number of profiles for each month (Fig. S1). The yearly and monthly
distributions of the temperature profiles lie along the middle of the Red Sea and are given in the
supplementary material (Tables S2-S3). As part of the quality check, 2679 profiles were removed from
the main dataset. A total of 2063 salinity profiles are available for the entire Red Sea (Table S4). MLD is
estimated based on the temperature profiles due to the increased number and sufficient monthly coverage
comparing to that of salinity. The spread of the temperature profiles used in this analysis is shown in Fig.
1.





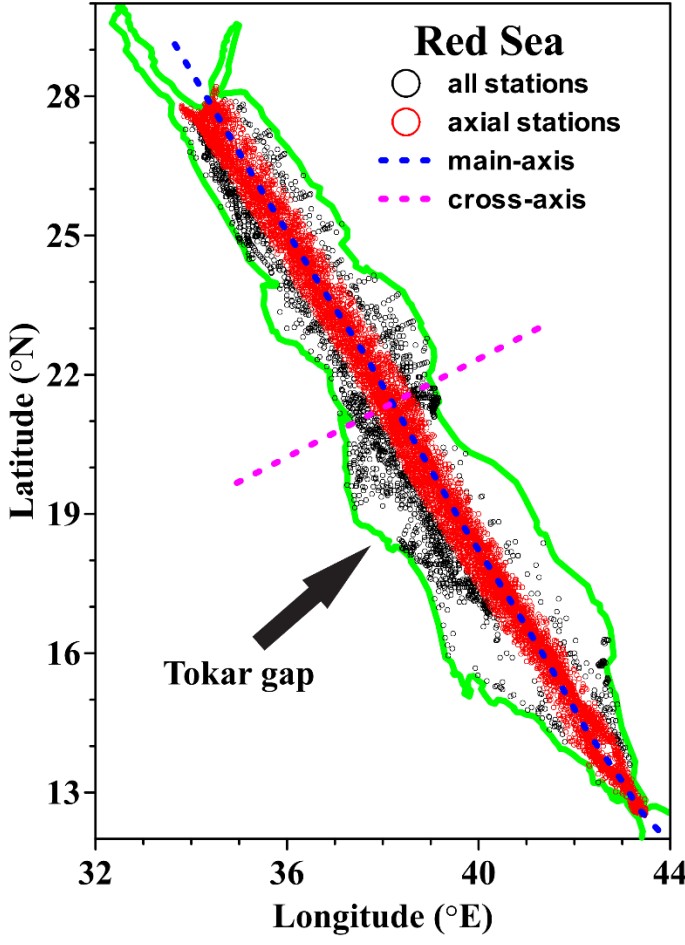

**Figure 1.** The locations of temperature profiles in the Red Sea. Black circles denote all available profiles, while red circles denote the profiles close to the main-axis that used for climatology calculation. The blue (magenta) dashed line indicate main-axis (cross-axis) of the Red Sea.

The monthly mean values of heat fluxes and wind stress data are provided by Tropflux at a 1°x1° spatial resolution for the period 1979-2016, which are used to check the influence on MLD variability (http://www.incois.gov.in/tropflux_datasets/data/monthly/). Tropflux captures better variability and less bias than the other available fluxes and wind stress products (Praveen Kumar et al., 2012, 2013). Since evaporation is not provided by Tropflux, the monthly mean values of evaporation from OAflux (from

1979        to        2016        and        1°x1°        spatial        resolution)        are        used

(ftp://ftp.whoi.edu/pub/science/oaflux/data_v3/monthly/evaporation/). The TRMM (Tropical rainfall measuring mission, https://pmm.nasa.gov/data-access/downloads/trmm) satellite provided the precipitation information for every 0.25°x0.25° grid and 3-hourly to monthly time scale from 1997 to 2016 (TRMM monthly 3B43_V7 product is used). Monthly climatology of heat flux, evaporation,

precipitation and wind stress are calculated. The period of precipitation data used for climatology calculation is shorter than other parameters. The present analysis is focusing on the seasonal timescale, and there shorter data period will not significantly affect the results.

The daily sea level anomaly (SLA) maps are provided by AVISO (www.aviso.oceanobs.com). These data are the merged product of satellite estimates from TOPEX/Poseidon, Jason-1, ERS-1/2, and Envisat and

are globally available for every 0.25°x0.25° grid from the year 1992 to present (Ducet et al., 2000; Traon and Dibarboure, 1999). The SLA maps are used to describe the eddy distribution in the Red Sea. Climate Forecast System Reanalysis (CFSR) provided hourly wind product from 1979 to 2010 at every 0.312°x0.312° grid (https://rda.ucar.edu/datasets/ds093.1/#!access) which is validated in the Red Sea by Aboobacker et al., (2016). CFSR hourly wind at 10 m above the surface is used to study the Tokar gap

winds.

## 2.2 Methods

Different approaches are available for MLD estimation. In the present study, MLD is estimated based on the recently introduced segment method, which is found to be less sensitive to short-range disturbances within the mixed layer (Abdulla et al., 2016). This method first identifies the portion of the profile

(segment) where the transition from a homogeneous layer to inhomogeneous layer occurs. Then, this segment is analyzed to determine the MLD.

The availability of profiles is denser along the middle of Red Sea during all months. The present analysis is performed for the profiles that fall within 0.5 degrees to the east and west of the main axis that, running along almost the middle of the Red Sea (hereafter called the "main axis"), has the advantage of a sufficient

number of profiles for every month. The main axis of the Red Sea is inclined to the west, with respect to





true north, by ~30 degrees. For this reason, instead of zonally averaging, the climatology is calculated by averaging the MLDs in an inclined direction parallel to the "cross-axis" (Fig. 1). The MLD is estimated for the individual profiles, and then, the monthly climatology is calculated every 0.5° from south to north (13° N to 27.5° N).

The heat flux, evaporation, precipitation and wind stress are interpolated to 0.5°x0.5° spatial grid to match with MLD climatology with the help of climate data operator (CDO) tool available at http://www.mpimet.mpg.de/cdo. The changes in surface water buoyancy forces is calculated following (Turner, 1973)

$$B_0 = \left(C_p^{-1}g \propto \rho_0^{-1}Q_{net}\right) + \left(-1 * g\beta s(E-P)\right) = B_{0T} + B_{0H} \tag{1}$$

where $C_p$ = water heat capacity, g = acceleration due to gravity, $\propto$=thermal expansion coefficient, $\rho_o$ = density of surface water, $Q_{net}$ = net heat flux at the sea surface, $\beta$ = haline contraction coefficient, s=salinity of surface water, E = evaporation rate, and P = precipitation. In Eq. (1), $B_{0T}$ and $B_{0H}$, respectively, represent the thermal and haline components of the buoyancy force. For ease of explanation, the Red Sea is divided into southern (13° N-18° N), central (18° N-23° N) and northern (23° N-28° N) regions and the seasons

defined as winter (Dec-Feb), spring (Mar-Apr), summer (May-Aug) and fall (Sep-Nov).

## 3 Results and discussion

### 3.1 MLD variability in the Red Sea

The Red Sea exhibits strong seasonal changes in its MLD, with deeper mixed layers during the winter and shallower ones during the summer, with gradual changes from deeper to shallower and vice versa in

the transitional months. A Hovmoller diagram of the monthly MLD climatology is presented in Fig. 2. The deepest MLD is observed in February; the shallowest, during May-Jun. A significant annual variability is observed along the Red Sea, with the largest amplitude of variability found at the northern tip of the Red Sea, with a maximum of MLD ~85 m (February) and a minimum of ~15 m (May-August). The northern Red Sea has often experienced deep convection during winter, on the order of 120 m to 150



m. Apart from the northern deep convection region, the south-central Red Sea (14° N-21° N) also experienced deeper MLDs during the winter, with an MLD of ~60 m. In contrast to the general pattern of shoaling during the summer, the region around ~19° N experienced a deeper mixed layer from July to September.

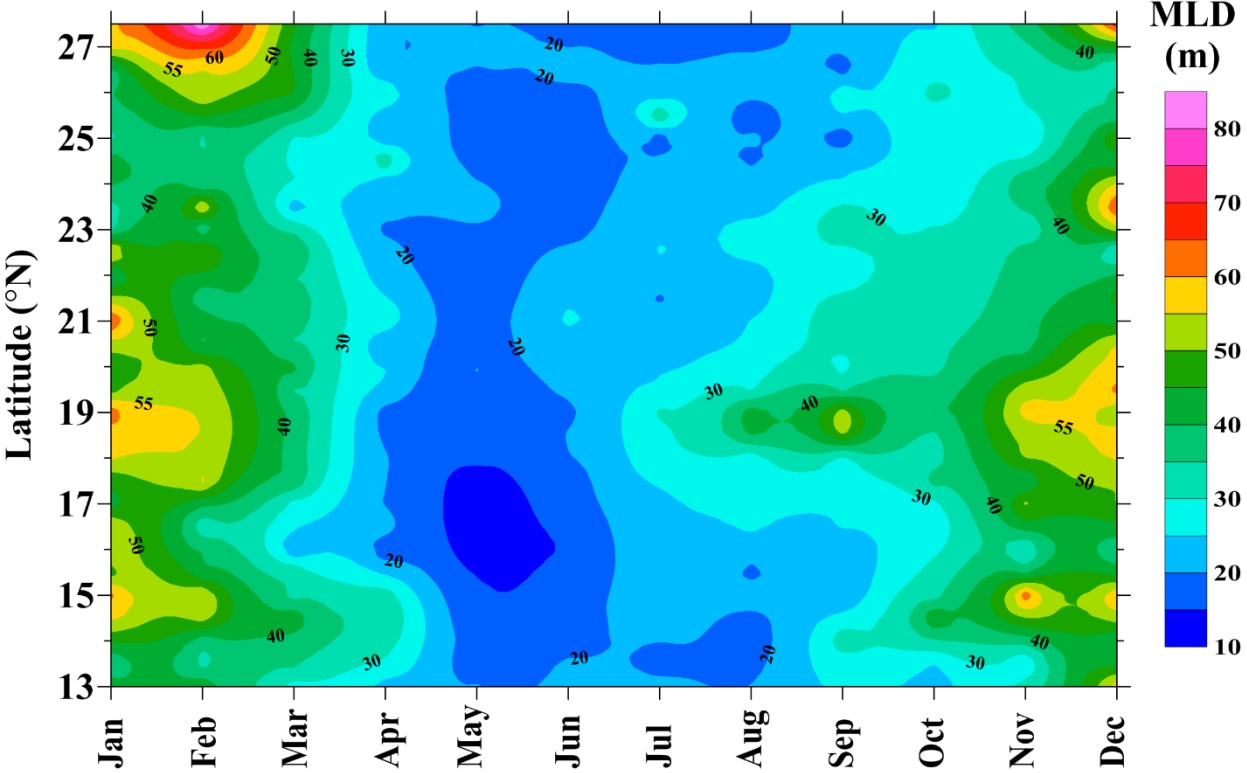

**Figure 2.** Hovmoller diagram of the MLD climatology along the axis of Red Sea.

The deepening of the MLD begins in October throughout the Red Sea. The winter cooling and its associated convection strengthen by December, with an average MLD>50 m. In this region, compared to other regions, in November and December, relatively shallower MLDs were witnessed at approximately 16° N-17° N, and 22° N-26.5° N. The winter deepening of the MLDs intensifies by January and continues

throughout February. In contrast to the general pattern of deeper MLDs in the northern latitudes, the area



between 24.5° N and 26.5° N shows a relatively shallow MLD almost throughout the year, especially in the winter.

The mixed layer starts to shoal gradually by the end of February, and the MLDs of most areas rise to ~20 m by April. Summer shoaling is comparatively stronger in the 15° N-18° N latitude band, and the detected mean MLD is < 15 m. Individual observations revealed that many profiles have MLDs < 5 m. In general, the shallow mixed layers are predominant from April to September, while this prevails until October in the far north. In the south-central Red Sea, the summer shallow mixed layer exists for only a short period, from April to June.

## 3.2 Major forces controlling the MLD variability

MLD is directly influenced by changes in the net heat flux (NHF), fresh-water flux (E-P) and wind stress. The different terms that contribute to NHF are given in Fig. 3 for a sample year 2016 in the central Red Sea. On an annual average basis, the incoming shortwave radiation (SWR, 202 W m$^{-2}$, positive downward) is mainly balanced by LHF (latent heat flux, -126 W m$^{-2}$) and LWR (long wave radiation, -83 W m$^{-2}$), while the SHF (sensible heat flux) is only -4 W m$^{-2}$. The net heat lose in the central Red Sea is 11 W m$^{-2}$. Both the LHF and LWR are gradually increasing towards the northern Red Sea. The monthly climatology of the NHF in the northern, central and southern Red Sea are given in Fig. 4a. Heat loss rises above 200 W m$^{-2}$ during December-January in the northern Red Sea, with a maximum of ~250 W m$^{-2}$ at the northern end of the sea in December. The annual mean of NHF is negative (heat loss) across the Red Sea, except for isolated locations in the southern Red Sea with trivial heat gain (figure not shown). The thermal components of the buoyancy forces calculated based on Eq. (1) show that the heat flux enhance mixing in the northern and central Red Sea during the winter, while it slightly diminishes mixing during summer. In the southern Red Sea, the effect of heat flux is relatively weak.




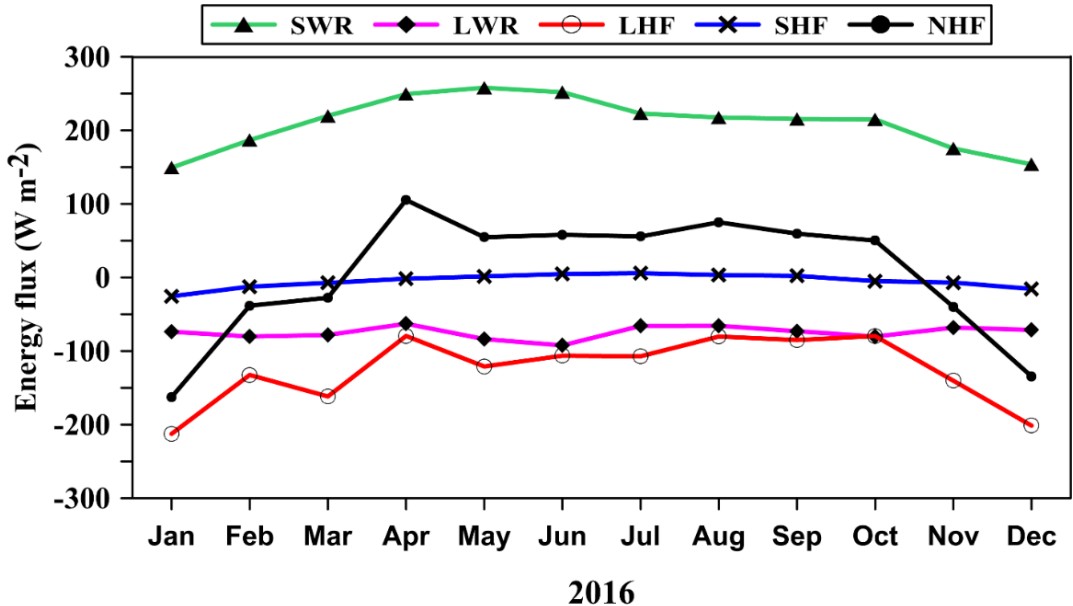

**Figure 3.** Time series of heat flux components for the year 2016 in the central Red Sea.

The evaporation rate in the Red Sea gradually increases from south to north (Fig. 4b). The central and northern Red Sea have higher evaporations during the winter (~6 mm day$^{-1}$) and moderate evaporations (~3 mm day$^{-1}$) during the summer. Evaporation shows weak seasonality in the southern Red Sea. Precipitation in the southern region is higher than those of the other areas of Red Sea, with maximum rainfall during July-September (Fig. 4b). The changes in buoyancy forces corresponding to fresh-water

flux (haline component) are estimated based on Eq. (1), which shows that the changes support mixing throughout the year and over the entire Red Sea. The thermal component is relatively higher than the haline component, and the net buoyancy follows a more or less similar pattern of thermal buoyancy all along the Red Sea (figure not shown). The observed variability of the above-discussed parameters is consistent with findings from earlier studies (Albarakati and Ahmad, 2013; Tragou et al., 1999).





**Figure 4.** Monthly climatology of a) NHF, b) evaporation and precipitation, and c) wind stress. South, central and north regions are represented by the changes at 14º N, 21º N and 27º N.

The pattern of wind stress in the Red Sea is significantly different from the other parameters. The wind stress is strong during the winter, leading to enhanced turbulence and mixing, while it is weak during the summer, resulting in a shallower mixed layer (Fig. 4c). Apart from that, strong surface winds blow to the Red Sea through the Tokar gap at approximately 19° N in July and August.



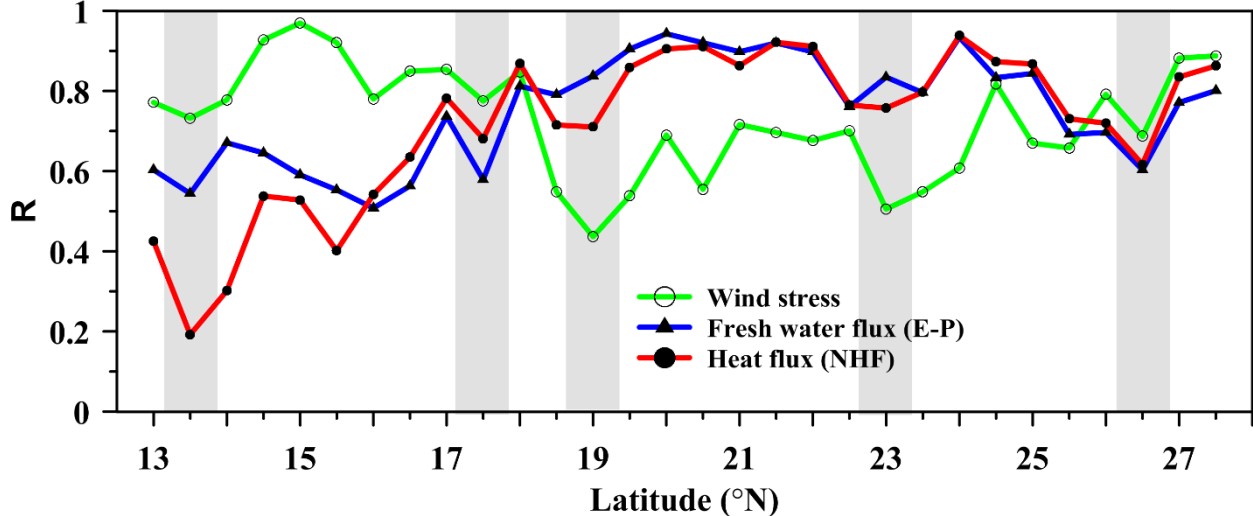

**Figure 5.** Correlation between major forces and MLD. Shaded regions represent locations of coinciding drops in correlation.

The correlations between MLDs and forcing factors are given in Fig. 5. Variability in MLD has an opposite phase to that of NHF, while the rest of the forcing are linearly phase related. NHF and E-P are well correlated (>0.8) with MLD in the central and northern Red Sea, and weakly correlated in the south. Wind stress has a higher correlation (>0.8) to the south, while it is relatively weakly correlated in the central and northern Red Sea. Toward the northern end, the wind stress gradually achieves a higher

correlation.

The results from Fig. 4 and 5 indicate that the MLD variability of the Red Sea is dominated by wind stress in the southern part, NHF (heatflux) and evaporation play a major role in the central region, while all the three are influencing in the northern region. Remarkably, for all the above-discussed parameters, coinciding drops are observed in the correlations at approximately 13.5° N, 17.5° N, 19° N, 23° N, and

26.5° N. These drops are discussed in the following section.





## 3.3 Impact of eddies

Satellite altimetry maps revealed the presence of a multiple eddies in the Red Sea which are often confined to specific latitude bands. Quadfasel and Baudner (1993) reported that most of the gyres in the Red Sea are concentrated in four latitude bands, approximately centered on 18° N, 20° N, 23° N and 26.5° N, and

some of these eddies are semi-permanent in nature. Johns et al. (1999) also reported presence of cyclonic eddies in the north and south of the Red Sea and anticyclonic eddies in the central Red Sea. Clifford et al. (1997) and Sofianos and Johns (2007) reported the presence of a quasi-permanent cyclonic gyre in the northern Red Sea during the winter. Analyzing the SLA maps from 1992 to 2012, Zhan et al., (2014) reported the presence of a multiple eddies with both polarities in the Red Sea. The number of identified

eddies peaked at approximately 19.5° N and 23.5° N. The upwelling proxy constructed using MODIS SST in the northern Red Sea shows the presence of frequent upwelling events at approximately 26.5° N almost every year (Papadopoulos et al., 2015) indicating presence of cyclonic eddy. The extent and time of the upwelling vary from year to year.

The eddy distribution in the Red Sea for the period from 1992-2012, based on SLA data is given in Fig.

6. The number of eddies are relatively higher in the central and northern Red Sea. The change in vertical stratification due to the presence of anticyclonic eddy (AE) and cyclonic eddy (CE) for different seasons are shown in Fig. 7. The black (green) colored curve represent the profile before (during) the eddy event. The date of profiling is given in the figure caption and the stations are marked. Figure 7a & 7f shows that the presence of AE during spring transformed the completely stratified upper layer to be well mixed till

50 m depth. Similar instance is shown in Fig. 7b & 7g where MLD changed from nearly zero to 30 m during summer. Figure 7c & 7h show the profiles corresponding to a CE event during fall, where shoaling of MLD by ~10 m is observed. Similarly, the CE event during winter lead to shoaling of mixed layer by ~60 m (Fig. 7d & 7i). Figure 7e & 7j show three profiles from single cruise collected within 12 hours which is coincided with the presence of CE and AE in a short distance, in which station A is located

outside the AE, B is located inside AE and C is partly in CE. There is a difference of ~100 m in the MLD due to the presence of eddies, in a short distance. Similarly, the MLD at station C is shallower than that of A due to the presence of a CE. The CE diminishes mixing through the upwelling of the subsurface





waters, while an AE enhances mixing through the downwelling of the surface waters. The impact of eddies varies based on the stratification and the amplitude and life time of eddies.

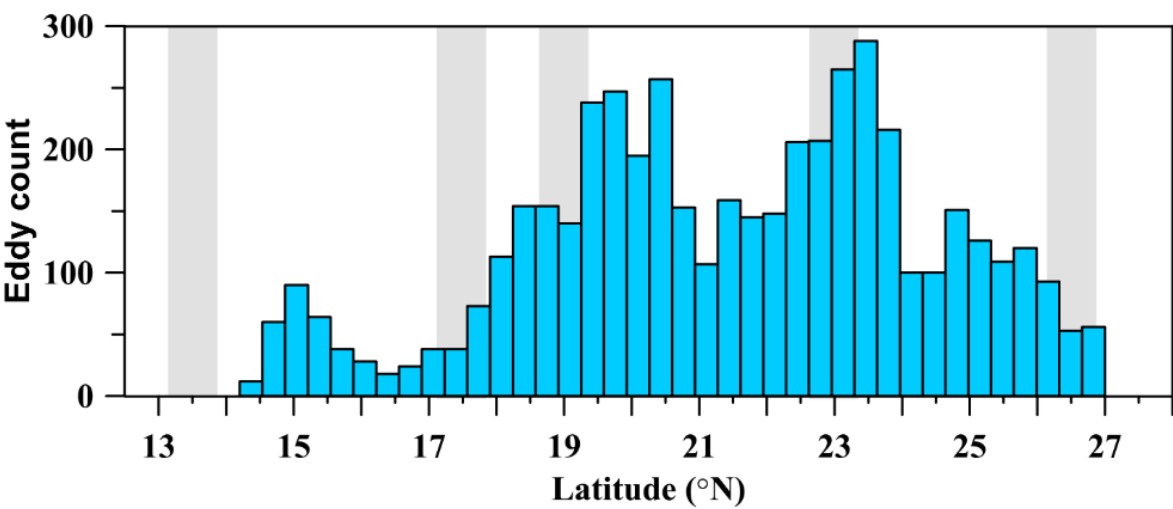


**Figure 6.** The number of eddies in the Red Sea derived from sea level anomaly for the period 1992-2012. The eddy count values are taken from Zhan et al., 2014. Shaded regions represent the location of correlation drops as shown in Fig. 5.



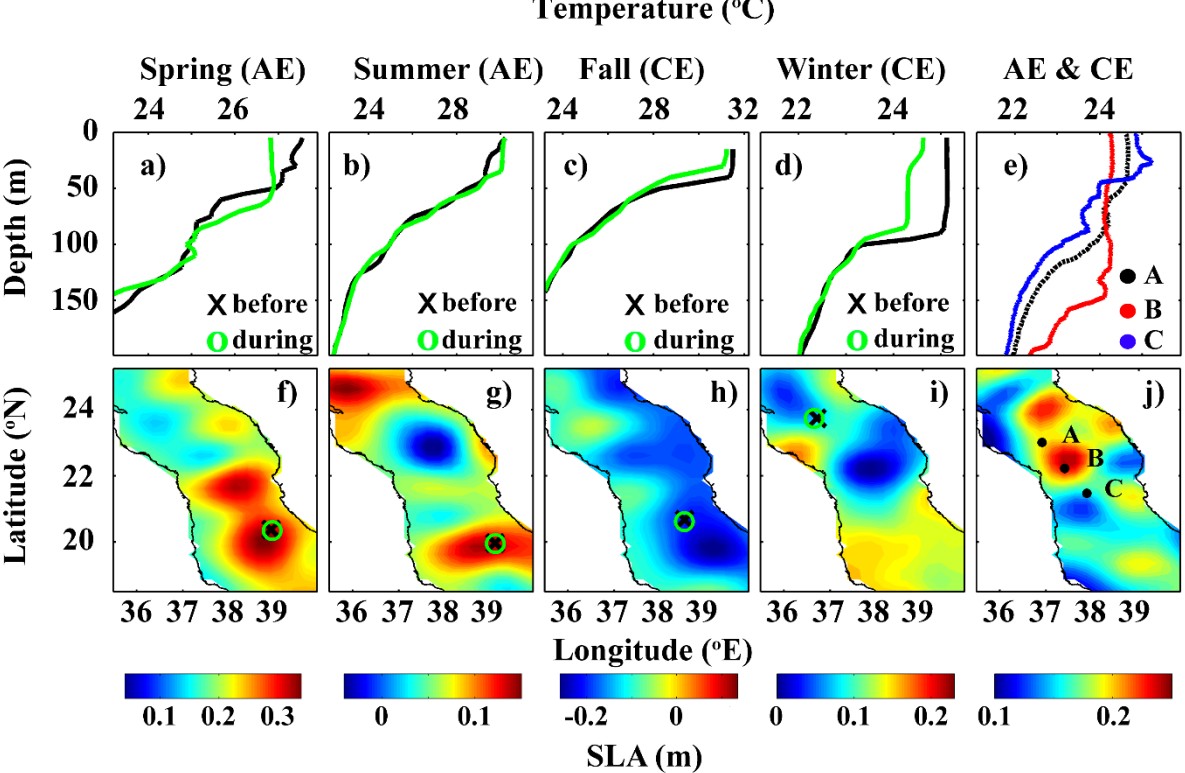

**Figure 7.** Profiles collected during (a) spring, (b) summer, (c) fall and (d) winter from the nearby stations in the Red Sea. The stations are marked on the SLA maps of the corresponding days (f-i). The "x" mark ("o" mark) represent profile collected before the appearance of the eddy (during the eddy period) and plotted in black (green) color. The dates of black and green profiles are respectively c) 11-03-2016 & 18-03-2016, e) 06-06-2016 & 13-06-2016, g) 16-09-2010 & 21-09-2010 and i) 13-12-2015 & 17-12-2015. The SLA is averaged for 5 days prior to the date of the later (green) profile. e) Temperature profiles collected from stations A, B & C within 12 hours (6th-7th Feb 2005) and j) the average SLA map for the period 4th to 7th Feb 2005.

The coinciding drops in the correlation curve, observed at approximately 19° N, 23° N and 26.5° N are well matching with the main eddy locations (Johns et al., 1999; Quadfasel and Baudner, 1993; Zhan et al., 2014), while those of 13.5° N and 17.5° N are not (Fig. 5 and 6). The Red Sea is very narrow at 13.5° N. Moreover, complex dynamics associated with the exchange of surface and subsurface waters between

the Red Sea and the Gulf of Aden occur in this region. The complexity of this region prevents linking the MLD variability directly to atmospheric forcing or eddies. The region at approximately 17.5° N is between the two eddy-driven downwelling zones at approximately 15° N and 19° N (Fig. 2). Mass

conservation requires upwelling to replace the downwelling water. The MLD climatology shows shallow mixed layers throughout the year at 17.5° N, which could be due to possible upwelling. Further investigation is required to unveil the dynamics associated with this region.

Rapid shoaling of the mixed layer is seen at ~26.5° N over a short distance (~100 km) adjacent to the deep convection zone in the northern side. The presence of a quasi-permanent cyclonic gyre during the

winter (Clifford et al., 1997; Sofianos and Johns, 2007) and frequent upwelling events (Papadopoulos et al., 2015) diminish the mixing in this region, leading to rapid shoaling of the mixed layer. The number of eddies has a minor peak at approximately 15° N. This region has a predominance of anticyclonic eddies (Zhan et al., 2014). The impact of the dominant anticyclonic eddies is visible in the MLD climatology, with deeper mixed layers at approximately 15° N (Fig. 2 and 7). The above results indicate that the

frequent eddies in the Red Sea significantly impact the MLD variability by enhancing/diminishing the mixing.

### 3.4 Influence of Tokar gap winds during the summer

The Tokar gap is one of the largest gaps in the high orography located on the African coast of the Red Sea, near 19° N. Strong winds are funneled to the Red Sea through this gap which last for few days to

weeks. Figure 8a shows the u-component of CFSR hourly surface wind at the Tokar region from 1996 to 2006. From the figure, it shows that the strong wind events occur during summer every year while the intensity and duration of the event varies from year to year. Tokar gap winds frequently attain a speed of 15 m s$^{-1}$. Previous research also show similar results (Jiang et al., 2009; Ralston et al., 2013; Zhai and Bower, 2013). Zhai and Bower (2013) reported that wind speed may reach 20 to 25m s$^{-1}$ based on ship

based observations. Figure 8b show that the onset of 2001 Tokar event was on 20$^{th}$ July and continued till 20$^{th}$ August, where the maximum wind speed occurred during this period compared to rest of the year. These strong winds generate strong turbulence in the surface water, which enhances vertical mixing.





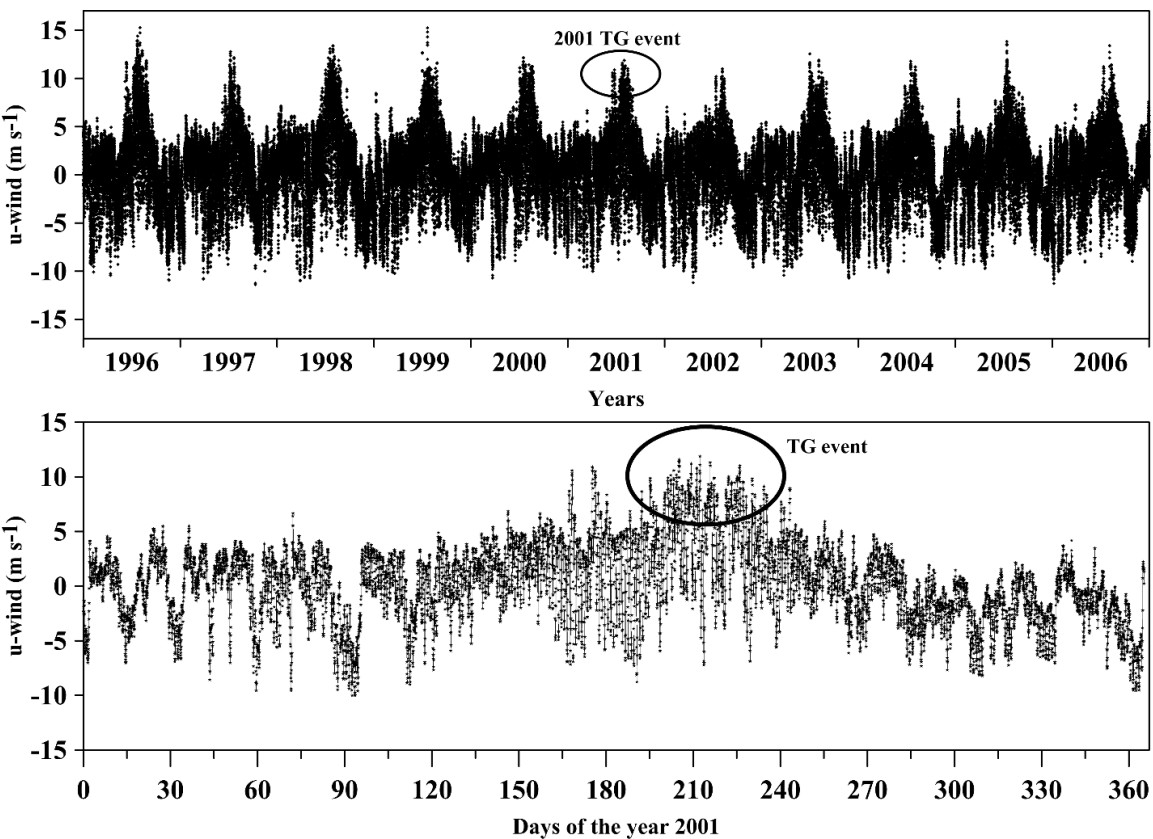

**Figure 8.** U-component of the hourly surface wind near the Tokar region (38.5° E, 18.5° N) a) from year
1996 to 2006 and b) for the year 2001. The ellipse indicates the TG event in the year 2001.

The temperature and salinity profiles measured during summer 2001 (13-14 Aug 2001), which coincided
with the Tokar event are shown (Fig. 9a-b). The signature of Tokar event is clearly visible in the satellite-
derived SLA, with well-defined cyclonic and anticyclonic eddies to the north and south of the Tokar gap
respectively (Fig. 9c-e). Both eddies have basin wide influence and radii between 70-80 kms.
Corresponding wind speed pattern (averaged for the previous 7 days) is shown (Fig. 9f-h). The profiles
to the north and south of the jet axis display significant difference in MLD, with a deeper mixed layer in
the south. Station A is far from both cyclonic and anticyclonic eddies and shows the expected MLD during
this period. The presence of the anticyclonic eddy at station B enhances strong downwelling, extending
the mixing to a depth approximately 80 m. It is to be noted that the entire Red Sea basin is well stratified



during this period, with MLDs ranging from 10 m to 15 m. Stations C and D are located at the edge of

the cyclonic eddy, and both have shallower thermocline and mixed layer.



**Figure 9.** (a) The CTD measured temperature and salinity profiles during 13-14 Aug 2001. (b) SLA maps
and (c) wind speed and direction (averaged for the previous one week) in the Tokar region, before, during
and after the Tokar event.

The MLDs of all the available profiles in the Tokar region before, during, just after and after a month of
the Tokar event are plotted in Fig. 10 (profiles for the first 15 days of each month are displayed). The
mean MLD, standard deviation and number of profiles are given in Table 1. Before the Tokar event, the
southern and northern sides of the Tokar axis (18° N-19.5° N and 19.5° N-21° N, respectively) displayed
similar mixed layers (Fig. 10a-c). During the Tokar event, the southern side experienced enhanced
mixing, while the northern side show shallow mixed layer (Fig. 10d-f).







**Figure 10**. Temperature profiles from the north of the Tokar axis (left panel, blue curves), south of the Tokar axis (middle panel, red curves) and the corresponding MLD (right panel) during the first 15 days of each month from July to October. The dashed line passes through 19.5° N, roughly separating the north and south of the Tokar axis. MLD of each profile is represented by the filled colors. The blue and red circles in (f) schematically represent cyclonic and anticyclonic eddies during Tokar event, respectively.

**Table 1.** The mean MLD in the north and south of Tokar jet axis from July to October.

| 1-15th days of the Month | Mean | | Standard deviation | | Number of profiles | |
|---|---|---|---|---|---|---|
| | North | South | north | south | north | south |
| Jul (before) | 20 | 26 | 5 | 8 | 19 | 12 |
| Aug (during) | 24 | 38 | 8 | 17 | 27 | 24 |
| Sep (just after) | 30 | 52 | 11 | 14 | 27 | 27 |
| Oct (after one month) | 31 | 34 | 9 | 12 | 36 | 30 |

The anticyclonic part of the Tokar induced eddies enhance downwelling and the associated deepening of the mixed layer along the southern side of the jet axis, while the cyclonic eddies generate upwelling and the associated shoaling of the mixed layer along the northern side. The profiles in September (just after

the Tokar event) show the southern side is well mixed by the event, which leads to an average difference
of 20 m in the MLDs between both sides of the Tokar axis (Fig. 10g-i). The signature of the Tokar events
in the MLDs has disappeared by October (one month after the Tokar event, Fig. 10j-l). The turbulence
induced by strong winds and associated eddies enhance vertical mixing at approximately 19° N during
the summer.

## 4 Conclusions

A detailed information on MLD variability is crucial for understanding the physical and biological
processes in the ocean. The goals of this study were to produce a climatology record of MLD for the Red
Sea and to investigate the role of major forces on MLD changes. With the help of in situ temperature
profiles from CTD, XBT, MBT and profiler float measurements, the MLD variability in the Red Sea has
been explored for the first time and the MLD climatology is produced for every 0.5 degrees along the
main axis. Averaging the MLDs can result in slightly lower values, but the climatology reasonably
captured all the major features. The present work provides a general picture of the MLD structure in the
Red Sea and its seasonal variability. Influences of wind stress, heat flux, evaporation and precipitation
are explored. Further, the impact of the Tokar gap jet stream winds, eddies and the upwelling events in
the northern Red Sea are also investigated.

A deep ventilation process associated with the winter cooling is observed across the entire Red Sea during
the months of December to February (Fig. 2). Similarly, very shallow MLDs associated with strong
stratification are detected all along the region from May to Jun. The climatological winter MLD ranges
from ~40 to 85 m (in January). Similarly, the climatological summer MLD varies from 10 to ~20 m (in
June), which may reach to >40 (in July). The mixed layer becomes deeper toward the north, even though
the pattern is not linear. The largest amplitude of variability is observed at the tip of the northern Red Sea,
and is associated with strong deep convection during the winter and shoaling during the summer. The
region at approximately 19° N experienced deeper than average MLD for most of the year. This region
experienced enhanced mixing during winter by surface cooling and during summer by Tokar gap winds.

The deepest mixed layer is observed at the northern tip of Red Sea during the winter, but the deep nature of northern mixed layer is almost limited to the winter months.

Correlation analyses between MLD and forcing factors displayed the influence of major forces on MLD, from north to south of the Red Sea. In general, the wind stress mainly controls the MLD variability in the southern part of the Red Sea, heat flux and evaporation dominate in the central region, and all the three forces contribute in the northern region. Coinciding drops are observed in the correlations for all the selected forcing factors around the previously reported main eddy locations. In these locations, eddies override the controls of the other main forces, namely, wind stress, heat flux and fresh-water flux. The quasi-permanent cyclonic gyre and upwelling in the northern Red Sea lead to the shoaling of the mixed layer at ~26.5° N throughout almost the whole year.

The Tokar gap winds during the summer enhanced the deep convection and mixing along the southern side of the Tokar jet axis. This leads to a deepening of the mixed layer, to >40 m, while the MLDs in the rest of the Red Sea are <20 m. The effect of Tokar event is seen in the profiles of late July to early August, and is gradually disappeared by October. The frequent eddies, associated with surface circulation and Tokar events, have strong impact on the MLD structure of the Red Sea.

**Data availability**

The climatology data produced in this manuscript is available from the repository "Figshare" (DOI:10.6084/m9.figshare.5539852). The monthly mean values of heat fluxes and wind stress data are available from Tropflux (http://www.incois.gov.in/tropflux_datasets/data/monthly/). The monthly mean values of evaporation is available from OAflux (ftp://ftp.whoi.edu/pub/science/oaflux/data_v3/monthly/evaporation/). The precipitation data is available from TRMM (https://pmm.nasa.gov/data-access/downloads/trmm).

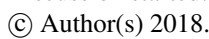



## Acknowledgments

This project was funded by the Deanship of Scientific Research (DSR), King Abdulaziz University, under grant number (438/150/129). The authors, therefore, acknowledge the DSR's technical and financial support. The authors acknowledge TropFlux, OAFlux, TRMM, AVISO, CFSR, World Ocean Database and Coriolis data center for making their data products publicly available. The authors also acknowledges the institutes who have provided CTD profiles from different cruises. The author CPA acknowledges the

Deanship of Graduate Studies, King Abdulaziz University, Jeddah, for providing a Ph.D. Fellowship.

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
