# Peer review of "Mixed layer depth variability in the Red Sea 1"

_Ocean Science, 2018_

## Short Comment (SC1) · 8 Mar 2018

The manuscript "Mixed layer depth variability in the Red Sea" discussed the variability by deriving the MLD monthly climatology from temperature profiles. This is probably the first time such a study has been conducted in the Red Sea. The paper is organised well and the discussions are relevant. It forms an important piece of information, especially on the MLD structure and the eddies in the Red Sea. This paper may be accepted in present form for publication in Ocean Science.

A minor comment:

The region around 18° N experiences the wind convergence during winter. Does it affect the MLD structure in the central Red Sea?

---

## Referee Comment (RC1) · Anonymous Referee #1 · 29 Mar 2018

General Comments: The authors have used historical temperature profiles from the Red Sea to develop a monthly climatology of mixed layer depth (MLD) along the seas' central axis, and investigate the importance of wind stress, thermal buoyancy forcing and haline buoyancy forcing in controlling MLD in the northern, central and southern Red Sea. The authors also investigate the relationship between MLD and the presence of cyclonic and anticyclonic mesoscale eddies, as well as the impact of the cross-axis Tokar Gap wind jet on MLD in summer. To my knowledge, this is the first published climatology of MLD in the Red Sea, which will be useful for verifying numerical model simulations of the basin and biogeochemical studies. The analysis of the impact of atmospheric forcing mechanisms as a function of latitude is interesting and worth publishing, in my view. The descriptions of the impacts of mesoscale eddies and the Tokar

[Figure]

Gap wind jet are less clear and I recommend major revisions in these sections to make them truly convincing.

Specific Comments:

Line 30 – Much of this is elementary physical oceanography, belonging in a textbook, not a scientific paper. I suggest to condense this part of the text significantly.

Line 43 – Bower and Farrar (2015) have direct estimates of evaporation rates that should be mentioned and referenced here.

Line 148 and following – 85 m +/- what? Need to add standard deviations to these mean values, in this line and all the following instances of reporting mean values. This is essential to understanding the statistical significance of the mean values. I realize some statistics are included in the supplementary material, but they should be included in the main document. Similar for lines 152–153, line 175 and line 197 (plotted lines need error bars or similar).

Line 214 – It wasn't obvious to this reviewer how the authors chose the latitudes where there were supposedly reductions in correlation between MLD and all forcing mechanisms. For some of the gray bars, the coincidence of lower correlations is obvious, but not in all. Would be good to define more clearly how these latitudes were chosen, hopefully using some objective criteria.

Line 243 – References are needed here to validate the authors' description of the relationship between mesoscale eddies and MLD.

Line 267 - The authors are implicitly arguing that the upwelling and downwelling associated with the secondary circulation of cyclonic and anticyclonic eddies is more important in determining MLD in the eddies than direct wind forcing and buoyancy forcing. Is there any literature to support this? I'm guessing there is, and the authors need to add some references here to this point.

Line 322 - It is not clear to me here if the region to the south of the jet axis is well-mixed

because of wind-induced turbulent mixing, or because of the secondary circulation associated with the wind stress curl-induced formation of the anticyclonic eddy, or both. the authors need to clarify this, or, if it is ambiguous, say that they are not sure which mechanism dominates.

Line 326 - If this is a summary sentence, I suggest to start it with "In summary..." I'm left with uncertainty about the authors' claim regarding the role of the TG jet in increasing MLD. As questioned above, does the upwelling and downwelling associated with the eddies overwhelm the direct mixing impact of the wind jet? Presumably the direct impact of the winds would be felt on both sides of the wind jet axis, but it's not clear if the authors are making this point for the cyclonic as well as anticyclonic eddy. Clarification needed here.

Line 346 – It was not clear to me if the deeper MLD was due to the direct impact of the winds or the formation of the anticyclonic eddy. Needs clarification.

Line 350 - I think this is the best result of the paper.

Line 357 - As remarked on above, why would the winds enhance ML development south of the wind axis but not north? Maybe the deepening to the south is due mostly/only to the formation of the anticyclonic eddy?

Technical comments:

There are numerous English grammatical and syntax errors in the writing. I've listed some, but not all, below.

Line 20 – Should read "The surface mixed layer ..." (i.e., add "the")

Line 30 – Should read "Oceanic heat loss...." (i.e., delete "The")

Line 30 – "to strong mixing..." "Strong"? Compared to what?

Line 35 – "through from….." Extra word here.

Line 39 – "The Red Sea is a typical....." How is it typical?

Line 45 – "regions in the world...." Referring to the water in the Red Sea? Maybe use "ocean basin" instead of "region."

Line 48 – What about Yao et al. references? Shouldn't they be included here? They represent some of the most comprehensive modeling studies of the Red Sea to date (after Sofianos' papers).

Line 49 – The increase in the number of temperature...." (add "the number of")

Line 55 – the authors should consider adding reference to Bower and Farrar (2015) paper and Yao et al. papers.

Line 77 – Over what depth range are inversions flagged?

Line 89 – What does "spread" mean? I think the word to be used is "distribution."

Line 110 – "Traon" Check spelling. I think it's "LaTraon".

Line 118 – What is meant by short-range disturbances? A sentence or two more on how the method works will save the reader from having to look it up elsewhere.

Line 121 – Would be helpful to the reader to give an example. How exactly does this work?

Line 149 – Are these numbers from individual profiles? Please clarify.

Line 158 – This sentence is confusing. What is meant by the "other regions"?

Line 168 – I would say April to June is more like the monsoon transition (probably low winds), not summer.

Line 174 – "net heat loss" (loss not lose)

Line 180–181 – Rather than "enhance mixing," which should be "enhances mixing" to be grammatically correct, I would suggest saying "supports vertical mixing through

buoyancy loss" or something similar.

Line 181 – "slightly diminishes mixing. . ." And here I would say "opposes vertical mixing due to buoyancy gain."

Line 184 – Would be helpful to define acronyms in figure caption.

Line 190 – "support vertical mixing" (add "vertical")

Line 192 – Shouldn't it be "net buoyancy flux" ?

Line 194 – Isn't there a Sofianos paper to be referenced here too?

Figures 3 and 4 – It would be helpful to add a zero line on Figs. 3 and 4.

Line 200 – I suggest that the authors mention the wind direction as well as stress amplitude variations through the seasons. Also, shouldn't wind stress in the winter be negative? All wind stress values are presented as positive. This is okay since it is only the magnitude (not direction) that impacts vertical mixing, but the authors need to say they are showing absolute value only.

Line 206 – I'm not sure what the authors mean here by "phase." I think they are referring to negative and positive correlation; e.g., MLD and NHF are negatively correlated since as NHF (into the ocean) increases, MLD decreases.

Line 229 - How were eddies identified?  If some eddies or sub-gyres are semi-permanent, how do you decide when one 'dies' and a new one is formed?  If the histogram is from another paper, it should be referenced here.

Line 258 - I think 'curve' should be 'curves,' because the point is (I think) that at these latitudes, correlations between MLD and all the forcing factors (wind, thermal buoyancy, haline buoyancy) are reduced.

Line 260 - Zhai and Bower 2013 should be added to this list, and Bower and Farrar 2015.

Line 289 – Authors should indicate data source in figure caption.

Line 291 – What is the data source for the T S profiles?

Line 293 - This year (2001) was also highlighted and described by Zhai and Bower 2013, which should be referenced here.

Line 333 - "slightly lower" than what? Lower than some individual measurements? If that is what is meant, that is obvious and this phrase should be deleted, or replaced with the actual extreme values.

Line 334 - Rather than 'general picture', authors should say something more concrete like 'climatological mean."

Line 340 - I would say that shallow MLD and increased stratification are the same thing. Authors could consider 'associated with increased short-wave radiation' instead.

Line 343 - Suggest to add "...is not linear with increasing latitude."

Line 345 - This phrase is confusing. Suggest to say "deeper MLD than typical of else-where in the Red Sea" or something similar.

REVIEWER COMMENTS – SUPPORTING INFORMATION

Figure S1 – I think "distribution" is a better word than "spread."

Table S1 – I don't think it's necessary to include this table. It was sufficient to describe the end result of the QC in the manuscript.

Table S2 – Could this information be summarized more efficiently with a plot of some kind?

Table S3 – Similar comment for this table. Change to a plot?

---

## Short Comment (SC2) · 13 Apr 2018

Response to the interactive comment on "Mixed layer depth variability in the Red Sea" by Cheriyeri P. Abdulla et al. (SC1).

The manuscript "Mixed layer depth variability in the Red Sea" discussed the variability by deriving the MLD monthly climatology from temperature profiles. This is probably the first time such a study has been conducted in the Red Sea. The paper is organised well and the discussions are relevant. It forms an important piece of information, especially on the MLD structure and the eddies in the Red Sea. This paper may be accepted in present form for publication in Ocean Science.

A minor comment:

[Figure]

The region around 18 °N experiences the wind convergence during winter. Does it affect the MLD structure in the central Red Sea?

Answer:

Thank you very much for your interest in the manuscript, and for your effort and time in reviewing. The convergence of wind is observed in the central Red Sea during winter, where NNE winds converge with SSW winds. This resulted in a relatively weak wind speed in the central Red Sea. The climatology of mixed layer in the central Red Sea during this period has shown relatively deeper mixed layer. It has been observed that the wind convergence during winter result pile up of sea level in the central Red Sea with maximum sea level around 19 °N. This is consistence with results from Sofianos et al., 2001. The pile-up of sea level and possible enhancement in vertical convection could be the reason for relatively deeper mixed layer in the central Red Sea.

---

## Author Comment (AC1) · 13 Apr 2018

*Anonymous Reviewer #1*

*General Comments:*

*The authors have used historical temperature profiles from the Red Sea to develop a monthly climatology of mixed layer depth (MLD) along the seas' central axis, and investigate the importance of wind stress, thermal buoyancy forcing and haline buoyancy forcing in controlling MLD in the northern, central and southern Red Sea. The authors also investigate the relationship between MLD and the presence of cyclonic and anticyclonic mesoscale eddies, as well as the impact of the cross-axis Tokar Gap wind jet on MLD in summer.*

*To my knowledge, this is the first published climatology of MLD in the Red Sea, which will be useful for verifying numerical model simulations of the basin and biogeochemical studies. The analysis of the impact of atmospheric forcing mechanisms as a function of latitude is interesting and worth publishing, in my view. The descriptions of the impacts of mesoscale eddies and the Tokar Gap wind jet are less clear and I recommend major revisions in these sections to make them truly convincing.*

*Answer:*

We thank the reviewer for his valuable comments and suggestions. The comments and suggestion were very helpful in improving the manuscript. Necessary improvements are done

in the sections explaining the effect of eddies and Tokar gap jet winds. The direct effect of wind apart from the effect of wind induced secondary circulation (the cyclonic and anticyclonic eddies) was not clear in the previous version of the manuscript, which is solved in the new version of manuscript. The answers to both specific and technical comments are given below, and required modifications are made in the manuscript.

Please note that coloured text is used in few instances of this document to represent modified/deleted text.

Red: the modified/deleted text in the previous version of the manuscript

Blue: the modified text in the new version of the manuscript.

**Specific Comments:**

*Specific Comment-1:*

*Line 30 – Much of this is elementary physical oceanography, belonging in a textbook, not a scientific paper. I suggest to condense this part of the text significantly.*

*Answer:*

We agree with the reviewer in this, our intension was to refresh the memories of the readers with the effect of each parameters on the mixed layer depth. As suggested by the reviewer, the paragraph is shortened as follows. In this, we show the two paragraphs before and after modification.

Previous version of the paragraph

Surface mixed layer is a striking and universal feature of the open ocean where the turbulence associated with various physical processes leads to the formation of a quasi-homogeneous layer with nearly uniform properties. The thickness of this layer, often named mixed layer

depth (MLD), is one of the most important oceanographic parameters, as this layer directly communicates and exchanges energy with the atmosphere and therefore has a strong impact on the distribution of heat (Chen, Busalacchi, & Rothstein, 1994), ocean biology (Polovina, Mitchum, & Evans, 1995) and near-surface acoustic propagation (Sutton, Worcester, Masters, Cornuelle, & Lynch, 2014). Heat and fresh-water exchanges at the air-sea interface and wind stress are the primary forces behind turbulent mixing. Similarly, stirring associated with turbulent eddies predominantly changes the mixing process, mainly along the isopycnal surfaces where stirring may occur with minimum energy (de Boyer Montégut, Madec, Fischer, Lazar, & Iudicone, 2004; Hausmann, McGillicuddy, & Marshall, 2017; Kara, Rochford, & Hurlburt, 2003).

Oceanic heat loss cools the mixed layer and weakens the stratification, leading to strong mixing and a deeper MLD. Similarly, the heat gain warms the mixed layer and strengthens the stratification, leading to weak mixing and shallow MLDs. The fresh-water loss makes the surface water more saline and denser, leading to enhanced mixing and deeper MLDs, while the fresh-water gain makes the surface water fresher and lighter, leading to diminished mixing and shallow MLDs. The momentum transmitted to the ocean through from the wind stress acts as the primary dynamic force for the upper layer turbulence and circulation. The shear and stirring generated by the wind stress enhance the vertical mixing and play a major role in controlling the deepening of the oceanic mixed layer. In some regions, the mixed layer variability is mainly controlled by wind stress.

New version of the paragraph (modified text in blue colour)

The surface mixed layer is a striking and universal feature of the open ocean where the turbulence associated with various physical processes leads to the formation of a quasi-homogeneous layer with nearly uniform properties. The thickness of this layer, often named mixed layer depth (MLD), is one of the most important oceanographic parameters, as this

layer directly communicates and exchanges energy with the atmosphere and therefore has a strong impact on the distribution of heat (Chen et al., 1994), ocean biology (Polovina et al., 1995) and near-surface acoustic propagation (Sutton et al., 2014). Heat and fresh-water exchanges at the air-sea interface and wind stress are the primary forces behind turbulent mixing. The loss of heat and/or freshwater from the ocean surface can weaken the stratification and enhance the mixing and vice versa. The shear and stirring generated by the wind stress enhance the vertical mixing and play a major role in controlling the deepening of the oceanic mixed layer. Further, the stirring associated with turbulent eddies predominantly changes the mixing process, mainly along the isopycnal surfaces where stirring may occur with minimum energy (de Boyer Montégut et al., 2004; Hausmann et al., 2017; Kara et al., 2003).

[Lines: 20-33]

*Specific Comment-2:*

*Line 43 – Bower and Farrar (2015) have direct estimates of evaporation rates that should be mentioned and referenced here.*

*Answer:*

> *The suggested reference of Bower and Farrar (2015) is appropriate to the context and added to the manuscript.*

> [Lines: 50]

*Specific Comment-3:*

*Line 148 and following – 85 m ± what? Need to add standard deviations to these mean values, in this line and all the following instances of reporting mean values. This is essential to understanding the statistical significance of the mean values. I realize some statistics are included in the supplementary material, but*

*they should be included in the main document. Similar for lines 152–153, line 175 and line 197 (plotted lines need error bars or similar).*

*Answer:*

The range of observed MLD values and the standard deviation from mean value are included at appropriate instances. The text is modified accordingly. The error-bars are added to the figure describing the monthly climatology of NHF, evaporation and precipitation, and wind stress (Figure 5).

The figure 4 shows monthly values from a single year (2016), with one value for each month. Therefore, we have not included the error bars.

Modified text

A Hovmoller diagram of the monthly MLD climatology is presented in Fig. 3. The deepest MLD is observed in February and the shallowest during May-Jun. A significant annual variability is observed in the Red Sea. The maximum value of climatological mean MLD is observed in February at the northern Red Sea while the minimum noticed at various instances, especially during summer months. The MLD of individual profiles in the northern Red Sea has a wide range of values from 40 to 120 m mainly due to the presence of active convection process, while some of the profiles show MLD deeper than 150 m in consistence with Yao et al., (2014). Apart from the northern deep convection region, the south-central Red Sea between 18 °N-21 °N (53±5 m) and 14 °N-16 °N (48±9 m) also experienced deeper MLDs during the winter, which is separated by a shallower MLD around 17 °N (44±14 m). During July to September, the region around 19 °N experienced a deeper mixed layer in contrast with the general pattern of summer shoaling over the entire Red Sea.

[Lines: 187-197]

*Specific Comment-4:*

*Line 214 – It wasn't obvious to this reviewer how the authors chose the latitudes where there were supposedly reductions in correlation between MLD and all forcing mechanisms. For some of the gray bars, the coincidence of lower correlations is obvious, but not in all. Would be good to define more clearly how these latitudes were chosen, hopefully using some objective criteria.*

*Answer:*

> As pointed out by the reviewer, we have selected the latitude bands (13.5 °N, 17.5 °N, 19 °N, 23 °N, and 26.5 °N) based on the observed drop in correlation for all the forces. We agree that there is a small difference in the case of 23 °N. At this latitude, the heat flux and wind stress have a clear coinciding drop in correlation while correlation for freshwater has a small increase. But, considering the correlation values for freshwater from 22 °N and 24 °N, the correlation is dropped around 23 °N (between 22 °N and 24 °N), even though a small local increase is seen at 23 °N. Therefore, we considered 23 °N as the region of coinciding drops in correlation.

> [Lines:293]

*Specific Comment-5:*

*Line 243 – References are needed here to validate the authors' description of the relationship between mesoscale eddies and MLD.*

*Answer:*

> Appropriate references were added to the text (Dewar, 1986; Fox-Kemper, Ferrari, & Hallberg, 2008; Hausmann et al., 2017; Smith & Marshall, 2009; de Boyer Montégut et al., 2004; Chelton et al., 2004, 2011).

> [Lines: 328-333]

*Specific Comment-6:*

*Line 267 - The authors are implicitly arguing that the upwelling and downwelling associated with the secondary circulation of cyclonic and anticyclonic eddies is more important in determining MLD in the eddies than direct wind forcing and buoyancy forcing. Is there any literature to support this? I'm guessing there is, and the authors need to add some references here to this point.*

*Answer:*

> As mentioned in the reply to comment #5, appropriate references discussing the importance and dominance of eddy effect on MLD variability are added to the manuscript. The results from the literature (de Boyer Montégut et al., 2004; Chelton et al., 2004, 2011; Dewar, 1986; Fox-Kemper et al., 2008; Hausmann et al., 2017; Smith & Marshall, 2009) have shown that eddies can efficiently re-stratify the ocean, dominating over the existing effect of wind stress and net heat flux over the region. The studies also show that the resultant effect of eddy is largely dependent on the eddy amplitude, and the mixing intensity is largest at the centre of eddy.

> [Lines: 329-333]

*Specific Comment-7:*

*Line 322 - It is not clear to me here if the region to the south of the jet axis is well-mixed because of wind-induced turbulent mixing, or because of the secondary circulation associated with the wind stress curl-induced formation of the anticyclonic eddy, or both. the authors need to clarify this, or, if it is ambiguous, say that they are not sure which mechanism dominates.*

*Answer:*

> It is true that the wind-induced turbulent mixing obviously exists on both side of the Tokar jet axis. The secondary circulation formed by the Tokar winds, with different polarities on

both side, cyclonic to the north of the Tokar-axis and anticyclonic to the south, acts in opposite direction to the vertical mixing. The mixing in the Tokar region is the sum of both the wind-induced turbulent mixing and the secondary circulation (eddies). A proper quantification of the contribution each mechanism needs further investigations.

[Lines: 439-443]

***Specific Comment-8:***

*Line 326 - If this is a summary sentence, I suggest to start it with "In summary. . ."*

***Answer:***

The text is changed accordingly.

[Lines: 443-444]

*I'm left with uncertainty about the authors' claim regarding the role of the TG jet in increasing MLD. As questioned above, does the upwelling and downwelling associated with the eddies overwhelm the direct mixing impact of the wind jet? Presumably the direct impact of the winds would be felt on both sides of the wind jet axis, but it's not clear if the authors are making this point for the cyclonic as well as anticyclonic eddy. Clarification needed here.*

***Answer:***

As mentioned in the reply to comment #7, we agree that the turbulence is present on both sides and enhances mixing. But eddy effect is in opposite directions in the northern and southern sides of Tokar-axis, and therefore the signature is evident in the mixed layer depth structure, with enhancement of mixing in the southern side and reducing the mixing in the northern side (please refer to Figures 10 and 11).

[Lines: 412, 424, 439-446]

*Specific Comment-9:*

*Line 346 – It was not clear to me if the deeper MLD was due to the direct impact of the winds or the formation of the anticyclonic eddy. Needs clarification.*

*Answer:*

> As mentioned in the reply to comment #7 and #8, the contribution of both the wind and secondary circulation are simultaneously existed in the Tokar region. In the previous version of the manuscript, the contribution of direct wind turbulence was not clearly mentioned in the conclusion part. In the revised version of the manuscript, we corrected the text and clearly stated that the mixing in the region is a combination effect of both wind turbulence and eddies.

> [Lines: 478-480]

*Specific Comment-10:*

*Line 350 - I think this is the best result of the paper.*

*Answer:*

> We thank the reviewer for appreciating this part of the result.

*Specific Comment-11:*

*Line 357 - As remarked on above, why would the winds enhance ML development south of the wind axis but not north? Maybe the deepening to the south is due mostly/only to the formation of the anticyclonic eddy?*

*Answer:*

> As stated in replies to comments#7 to 9, the wind enhance mixing on both side of the Tokar-axis. The text also corrected accordingly.

The deepening in the south of Tokar axis is the combined effect of both wind and anticyclonic eddy, while shoaling in north is due to the opposite (diminishing) effect of the cyclonic eddy.

[Lines: 490-492]

**Technical comments:**

*Technical comment-1:*

*Line 20 – Should read "The surface mixed layer . . ." (i.e., add "the")*

*Answer:*

The manuscript is corrected as suggested.

[Lines: 20]

*Technical comment-2 to 4:*

*Line 30 – Should read "Oceanic heat loss. . .." (i.e., delete "The")*

*Line 30 – "to strong mixing. . ." "Strong"? Compared to what?*

*Line 35 – "through from. . ..." Extra word here.*

**Answer to *Technical comment-2 to 4*:**

The manuscript is corrected accordingly. This paragraph is modified and summarised.

[Lines: 27-30]

*Technical comment-5:*

*Line 39 – "The Red Sea is a typical. . ..." How is it typical?*

*Answer:*

The text is corrected and removed the usage "typical".

We have used this word considering that the Red Sea is a typical inverse estuarine system, where the evaporation is dominated over the precipitation.

[Lines: 34]

*Technical comment-6:*

*Line 45 – "regions in the world. . .." Referring to the water in the Red Sea? Maybe use "ocean basin" instead of "region."*

*Answer:*

The manuscript is corrected as suggested.

[Lines: 40]

*Technical comment-7:*

*Line 48 – What about Yao et al. references? Shouldn't they be included here? They represent some of the most comprehensive modeling studies of the Red Sea to date (after Sofianos' papers).*

*Answer:*

The manuscript is corrected as suggested and the references are added.

[Lines:42-43]

*Technical comment-8:*

*Line 49 – The increase in the number of temperature. . .." (add "the number of")*

*Answer:*

The manuscript is corrected as suggested.

[Lines: 44]

*Technical comment-9:*

*Line 55 – the authors should consider adding reference to Bower and Farrar (2015) paper and Yao et al. papers.*

*Answer:*

The manuscript is corrected as suggested and the references *are added.*

[Lines: 49-50]

***Technical comment-10:***

*Line 77 – Over what depth range are inversions flagged?*

*Answer:*

Over upper 500 meters, which could be sufficient for the MLD estimation in the Red Sea.

***Technical comment-11:***

*Line 89 – What does "spread" mean? I think the word to be used is "distribution."*

*Answer:*

The text corrected as suggested.

[Lines: 83]

***Technical comment-12:***

*Line 110 – "Traon" Check spelling. I think it's "LaTraon".*

*Answer:*

The text corrected as suggested.

[Lines: 113]

***Technical comment-13:***

*Line 118 – What is meant by short-range disturbances? A sentence or two more on how the method works will save the reader from having to look it up elsewhere.*

*Line 121 – Would be helpful to the reader to give an example. How exactly does this work?*

*Answer:*

A brief description on the estimation of MLD using "segment method" is added in the manuscript, with the help of a sample profile.

The text added in the manuscript:

The MLD can be estimated based on different methods. The Fig.2 shows a sample temperature profile collected on 19th January 2015 from Red Sea (24.9° N, 35.18 °E), with short-range gradients within the mixed layer. This gradient could rise from instrumental errors or turbulence in the upper layer. The curvature method (Lorbacher et al., 2006) identified MLD at 32 m, due to the presence of a short range gradient at this depth. Threshold method (de Boyer Montégut et al., 2004) detected MLD at 130 m (threshold = 0.2 °C), while segment method identified MLD at 120 m. The segment method based MLD could be considered as a reliable estimate comparing to both curvature (under estimation) and threshold method (over estimation). The segment method first identifies the portion of the profile with significant inhomogeneity where the transition from a homogeneous layer to inhomogeneous layer occurs. Then, this portion of the profile is analyzed to determine the MLD (detailed procedure of the estimation technique is given Abdulla et al., 2016). In the present study, MLD is estimated based on the segment method, which is found to be less sensitive to short-range disturbances within the mixed layer (Abdulla et al., 2016). This method first identifies the portion of the profile (segment) where the transition from a homogeneous layer to inhomogeneous layer occurs. Then, this segment is analyzed to determine the MLD.

[Figure]

**Figure.** The MLD estimated for a schematic temperature profile based on curvature, threshold, and segment methods. Z-top and Z-bot represents the top and bottom ends of the portion of the profile with significant inhomogeneity.

[Lines: 119-161]

*Technical comment-14:*

*Line 149 – Are these numbers from individual profiles? Please clarify.*

*Answer:*

Yes, these numbers are from individual profiles. The same is mentioned in the text also.
The text from the manuscript

The MLD of individual profiles in the northern Red Sea has a wide range of values from 40 to 120 m mainly due to the presence of active convection process, while some of the profiles show MLD deeper than 150 m in consistence with Yao et al., (2014).
[Lines: 191-193]

*Technical comment-15:*

*Line 158 – This sentence is confusing. What is meant by the "other regions"?*

*Answer:*

The sentence is corrected. "other regions" is replaced with "other parts of the Red Sea"
The text from the manuscript

Compared to other parts of the Red Sea, during November and December, relatively shallower MLDs were witnessed at approximately 16 °N-17 °N, and 24.5 °N-26.5 °N.
[Lines: 218-220]

*Technical comment-16:*

*Line 168 – I would say April to June is more like the monsoon transition (probably low winds), not summer.*

*Answer:*

The text is corrected accordingly.
[Lines: 229-230]

*Technical comment-17:*

*Line 174 – "net heat loss" (loss not lose)*

*Answer:*

      The text is corrected accordingly.

      [Lines: 236]

*Technical comment-18:*

*Line 180–181 – Rather than "enhance mixing," which should be "enhances mixing" to be grammatically correct, I would suggest saying "supports vertical mixing through buoyancy loss" or something similar.*

*Answer:*

      The text is corrected accordingly

      [Lines: 242-243]

*Technical comment-19:*

*Line 181 – "slightly diminishes mixing. . ." And here I would say "opposes vertical mixing due to buoyancy gain."*

*Answer:*

      The text is corrected accordingly

      [Lines: 243-244]

*Technical comment-20:*

*Line 184 – Would be helpful to define acronyms in figure caption.*

*Answer:*

      The figure caption is corrected accordingly

**Figure 3.** Time series of heat flux components (incoming shortwave radiation (SWR), long wave radiation (LWR), latent heat flux (LHF), sensible heat flux (SHF) and net heat flux (NHF)) for the year 2016 in the central Red Sea.

[Lines: 251-253]

*Technical comment-21:*

*Line 190 – "support vertical mixing" (add "vertical")*

*Answer:*

*The text is corrected as suggested.*
[Lines: 204]

*Technical comment-22:*

*Line 192 – Shouldn't it be "net buoyancy flux" ?*

*Answer:*

The text corrected accordingly.
[Lines: 259]

*Technical comment-23:*

*Line 194 – Isn't there a Sofianos paper to be referenced here too? Figures 3 and 4 – It would be helpful to add a zero Line on Figs. 3 and 4.*

*Answer:*

The reference of *Sofianos paper* is included accordingly.

[Lines: 264-265]

The zero lines are inserted in the figure 3 and 4

[Lines: 250, 266]

*Technical comment-24:*

*Line 200 – I suggest that the authors mention the wind direction as well as stress amplitude variations through the seasons. Also, shouldn't wind stress in the winter be negative? All wind stress values are presented as positive. This is okay since it is only the magnitude (not direction) that impacts vertical mixing, but the authors need to say they are showing absolute value only.*

*Answer:*

The Figure in the previous version of manuscript show the "magnitude of wind stress" alone. As mentioned by the reviewer, the East and North components of wind stress along with absolute wind stress are presented in the figure 5.

[Lines:266]

*Technical comment-25:*

*Line 206 – I'm not sure what the authors mean here by "phase." I think they are referring to negative and positive correlation; e.g., MLD and NHF are negatively correlated since as NHF (into the ocean) increases, MLD decreases.*

*Answer:*

The text is corrected accordingly as follows.

The wind stress and E-P are positively correlated with MLD while the NHF is negatively correlated since as NHF (into the ocean) increases, MLD decreases. For simplicity of the figure (Figure 5), the correlation values of all parameters are presented as positive.

[Lines: 279-282]

*Technical comment-26:*

*Line 229 - How were eddies identified? If some eddies or sub-gyres are semi-permanent, how do you decide when one 'dies' and a new one is formed? If the histogram is from another paper, it should be referenced here.*

*Answer:*

> Eddies are identified based on "winding angle" method. The identification is done by Zhan et al., 2014. The reference is mentioned in the text as well as in the caption of the histogram (Fig. 7).
>
> [Lines: 308-309 and 335]

*Technical comment-27:*

*Line 258 - I think 'curve' should be 'curves,' because the point is (I think) that at these latitudes, correlations between MLD and all the forcing factors (wind, thermal buoyancy, haline buoyancy) are reduced.*

*Answer:*

> The text is corrected accordingly
>
> [Lines: 362]

*Technical comment-28:*

*Line 260 - Zhai and Bower 2013 should be added to this list, and Bower and Farrar 2015.*

*Answer:*

> These references are added to the list.
>
> [Lines: 364]

*Technical comment-29:*

*Line 289 – Authors should indicate data source in figure caption.*

*Answer:*

The wind data is CFSR hourly wind product. The same mentioned in the caption also.

[Lines: 394]

*Technical comment-30:*

*Line 291 – What is the data source for the T S profiles?*

*Answer:*

The temperature and salinity profiles are from Sofianos and Johns, 2007, and the same is mentioned in the figure cation.

Caption

**Figure 9.** (a) The CTD measured temperature and salinity profiles during 13-14 Aug 2001. (b) SLA maps and (c) wind speed and direction (averaged for the previous one week) in the Tokar region, before, during and after the Tokar event. The temperature and salinity profiles are received through personal communication from (Sofianos & Johns, 2007).

[Lines: 412-416]

*Technical comment-31:*

*Line 293 - This year (2001) was also highlighted and described by Zhai and Bower 2013, which should be referenced here.*

*Answer:*

The suggested reference is added in the manuscript.

Text from the manuscript

The temperature and salinity profiles measured during summer 2001 (13-14 Aug 2001), which coincided with the Tokar event are shown in Fig. 9a-b (Sofianos and Johns, 2007; Zhai and Bower, 2013).

[Lines: 397-398]

*Technical comment-32:*

*Line 333 - "slightly lower" than what? Lower than some individual measurements? If that is what is meant, that is obvious and this phrase should be deleted, or replaced with the actual extreme values.*

*Answer:*

The text is corrected accordingly.

[Lines:453]

*Technical comment-33:*

*Line 334 - Rather than 'general picture', authors should say something more concrete like 'climatological mean."*

*Answer:*

The text is corrected accordingly.

[Lines: 454]

*Technical comment-34:*

*Line 340 - I would say that shallow MLD and increased stratification are the same thing. Authors could consider 'associated with increased short-wave radiation' instead.*

*Answer:*

The text is corrected accordingly.

[Lines: 459-460]

*Technical comment-35:*

*Line 343 - Suggest to add "...is not linear with increasing latitude."*

*Answer:*

The text is corrected accordingly.

[Lines: 475]

***Technical comment-36:***

*Line 345 - This phrase is confusing. Suggest to say "deeper MLD than typical of elsewhere in the Red Sea" or something similar.*

*Answer:*

The text is corrected accordingly.

[Lines: 478]

**Answer to reviewer comments-Supporting information**

*SI-Comment-1:*

*Figure S1 – I think "distribution" is a better word than "spread."*

*Answer:*

The text is corrected accordingly.

[Lines: 27 and 45]

*SI-Comment-2:*

*Table S1 – I don't think it's necessary to include this table. It was sufficient to describe the end result of the QC in the manuscript.*

*Answer:*

*This Table is removed. The manuscript is modified accordingly.*

*SI-Comment-3 and 4:*

*Table S2 – Could this information be summarized more efficiently with a plot of some kind?*

*Table S3 – Similar comment for this table. Change to a plot?*

*Answer:*

As suggested by the reviewer, the tables S2, S3 and S4 are converted into plots.

[Lines: 52, 61, and 70]

---

## Author Comment (AC3) · 13 Apr 2018

Supporting Information for

**Mixed layer depth variability in the Red Sea**

Cheriyeri P. Abdulla[1*], Mohammed A. Alsaafani[1,2], Turki M. Alraddadi[1], and Alaa M. Albarakati[1]

[1]Department of Marine Physics, Faculty of Marine Sciences, King Abdulaziz University, Jeddah, Saudi Arabia.
[2]Department of Earth & Environmental Sciences, Faculty of Science, Sana'a University, Yemen.

*Correspondence to*: Cheriyeri P. Abdulla (acp@stu.kau.edu.sa)

**Contents of this file**

<#>Tables S1¶
<#>Tables S2¶
<#>Tables S3¶
<#>Tables S4¶

**Introduction**

The figures and tables which support the results discussed in the manuscript "**Mixed layer depth variability in the Red Sea**" are given below. Detailed description on the attached figures and tables is given in the manuscript. A brief summary is given in the following lines.

Figure S1 shows the monthly distribution of the temperature profiles along the main-axis of Red Sea. The number of profiles are marked at the top right corner in each sub-figure. This document also includes bar charts showing the number of selected profiles along the main-axis of Red Sea based on the year (Fig. S2) and month (Fig. S3) of measurement. Similarly, a bar chart is included to show the number of profiles with salinity values in the study area (Fig. S4).

[Figure]

**Figure S1.** Monthly distribution of the temperature profiles along the main-axis of Red Sea. The number of profiles is marked at the top right corner. The different steps of quality control are discussed in the main manuscript.

[Figure]

**Figure S2.** Number of temperature profiles along the main-axis of Red Sea based on the year of measurement.

[Figure]

**Figure S3.** Number of temperature profiles along the main-axis of Red Sea based on the month of
measurement.

[Figure]

**Figure S4.** Number of total salinity profiles available in the entire Red Sea based on the year of measurement.

---

## Referee Comment (RC2) · Anonymous Referee #2 · 1 May 2018

General comments:

The paper is generally well-executed and well-written, although it does need some editing. The authors say that MLD results are not available from previous research, but I am not competent to judge that, so I will take them at their word on that point. There is a lot of good material here, but I also have reservations about some of it. These comments are summarized below. Please note that these comments are not in order of importance, but are in the order I encountered the material in the paper.

The bottom line is that the basic description is well-done and should be published. The special latitude bands identified in the correlation plot are not proven to be real, at least to my satisfaction, but the Tokar Gap signal at 19N is interesting and corresponds to a clear "tongue" in the MLD climatology. I'm ignoring a rule I agree with that we cannot

just point to features in a plot and interpret these without a "null" test that the feature could be noise, but we'll discuss that more down below. The paper would be much better if you were to get rid of the AVISO SLA analysis and Section 3.3 and the other latitude bands and focus on the overall description and the Tokar Gap results, again see discussion below.

I should say that after writing this review I read the comments by the first anonymous reviewer. This person gives a very thorough review, and we have points of agreement and disagreement. I think the major disagreement is how we view the material concerning the Tokar Gap winds and subsequent eddy spin-up. I really liked this material, but the first reviewer perhaps did not like it so much. I think this is for the authors and the editor to sort out.

Specific comments:

L40 – I am not sure that "deep water formation" is appropriate. Common usage of that term is for NADW and ABW. Perhaps "intermediate water formation"? At the least tell us how deep this high salinity water reaches.

L95 – 1 by 1 degree spacing is very coarse for this region. With such a model can you really expect to resolve the scales that are important in the Red Sea?

L108-115 – The AVISO SLA is HIGHLY suspect in the Red Sea for resolving eddies. Yes, they grid it at quarter degree spacing, but how much actual data is there? Also, their covariance functions for the OI are not tuned to the Red Sea in general, or to the Tokar Gap in particular. See below, too, but I would remove the AVISO SLA eddy material and focus on the Tokar Gap analysis that doesn't require it.

L122-129 – Well-done to switch to an along-axis coordinate system and to do the analysis in an along-axis, time space. Nice!

L150-162 – I'm worried that we are over-interpreting in this section. The general seasonal pattern in clear, but the along-axis changes are not so clear. Without error bars

it is difficult to tell when we are interpreting real changes, or just noise.

L160 – As an example of the comment I just made, I cannot make any sense of a "general pattern of deeper MLDs in the latitudes". I just cannot see in Figure 2 where this statement comes from! You will need to be much more specific to convince me of this and it will also require appropriate error estimates. You lose little by sticking to the bigger picture and skipping the "wiggles", although the "tongue" at the latitude of the Tokar Gap is interesting (see below).

L205 and Figure 5 – I like this plot, but some estimate of the degrees of freedom needs to be made so that we know whether the structure in the curve is real or just statistical fluctuations. And, as usual, the degrees of freedom estimate should consider red noise and not assume white, or independent, noise.

L211-215 – I think we are over-interpreting again. These are very small "wiggles" and 2 of the 5 do not even show the pattern you assert, meaning all three curves moving down together. Again, if you want to interpret these small changes, then you have to do a much more thorough job on the statistics to convince us that we're not just looking at random fluctuations.

L216, all of Section 3.3 – As I mentioned above the analysis using the AVISO SLA is highly questionable here. I think this entire section, and basically all use of AVISO, is not necessary for this paper. You have some good results, but by pushing too far you run the risk of most readers doubting everything. Please note that I am trying to be constructive here and help to improve the paper. I like the paper as a whole, but really do not like this section.

L258-260, Figure 6 – This is a continuation of the previous comment. You say that there is a good match with the number of eddies and the latitude ranges you identified earlier from the correlation curves, but I simply do not see that. And since we're just doing the analysis "by eye", then my eye is as good as yours. You really have to do some statistics if you want to make this point. And once again, none of these things

are core results of your paper.

L277, all of Section 3.4 – Much better! Looking at the 19N signal where the Tokar Gap is, and showing the actual wind results rather than AVISO eddy counts is much more convincing. And note that this is the only latitude band where I see convincing results. It's well-known that strong winds through mountain gaps generate the eddy signals you infer (search for results on the Hawaiian Ridge and the Gulf of Tehuantepec), so you do not need the questionable AVISO results in order to rationalize the "tongue" in Figure 2 at 19N. This is a very nice result.
* * *

---

## Author Comment (AC4) · 23 May 2018

**Referee #2**

**General comments:**

The paper is generally well-executed and well-written, although it does need some editing. The authors say that MLD results are not available from previous research, but I am not competent to judge that, so I will take them at their word on that point.

There is a lot of good material here, but I also have reservations about some of it. These comments are summarized below. Please note that these comments are not in order of importance, but are in the order I encountered the material in the paper.

The bottom line is that the basic description is well-done and should be published.

The special latitude bands identified in the correlation plot are not proven to be real, at least to my satisfaction, but the Tokar Gap signal at 19N is interesting and corresponds to a clear "tongue" in the MLD climatology.

I'm ignoring a rule I agree with that we cannot just point to features in a plot and interpret these without a "null" test that the feature could be noise, but we'll discuss that more down below.

The paper would be much better if you were to get rid of the AVISO SLA analysis and Section 3.3 and the other latitude bands and focus on the overall description and the Tokar Gap results, again see discussion below.

I should say that after writing this review I read the comments by the first anonymous reviewer. This person gives a very thorough review, and we have points of agreement and disagreement. I think the major disagreement is how we view the material concerning the Tokar Gap winds and

subsequent eddy spin-up. I really liked this material, but the first reviewer perhaps did not like it so much. I think this is for the authors and the editor to sort out.

Reply:

Thank you very much for your precious comments and suggestions on the manuscript. They were very helpful in improving the manuscript. The reply to specific comments are given below and the manuscript is modified accordingly.

**Specific comments:**

**SC#1:**

L40 – I am not sure that "deep water formation" is appropriate. Common usage of that term is for NADW and ABW. Perhaps "intermediate water formation"? At the least tell us how deep this high salinity water reaches.

Reply:

The RSOW is an intermediate water formed in the northern part of Red Sea as part of the convection activity, which propagate through Bab-el-Mandab strait to the Gulf of Aden (Alsaafani & Shenoi 2007) and later spreads to the Indian Ocean, whose signature reaches into the south Indian Ocean about 6000 km away from the source (Beal et al., 2000). The sentence is corrected accordingly.

The earlier text in the manuscript:

It is one of the important deep water formation regions, and its signature reaches into the Indian Ocean (Beal et al., 2000).

The modified text in the manuscript:

It is one of the important intermediate water formation regions in the world (Red Sea Outflow Water, RSOW), formed mainly due to the open ocean convection in the northern Red Sea [Sofianos and Johns, 2003], which propagates through Bab-el-Mandab to the Gulf of Aden (Alsaafani and Shenoi 2007) and later spreads to the Indian Ocean, whose signature reaches into the south Indian Ocean  about 6000 km away from the source (Beal et al., 2000).

**SC#2:**

L95 – 1 by 1 degree spacing is very coarse for this region. With such a model can you really expect to resolve the scales that are important in the Red Sea?

Reply:

We have crosschecked the estimates from reanalysis flux products (Tropflux and OAflux) with previous studies in the Red Sea, and found that the variability are consistent with observations (Sofianos and Johns 2003, Murray and Johns 1997, Tragou et al. 1999, Sofianos et al. 2002, Farrar et al. 2009 and Yu and Weller 2007). Further, the variability of the flux parameters (Net heat flux and evaporation. Precipitation is negligible) along the main axis of the Red Sea is relatively smooth, and the general variability can be captured with 1 by 1 degree spacing. Therefore, the Tropflux and the OAflux estimates can be used to understand general variability of these parameters.

In the case of wind, which vary relatively rapid comparing to heat and fresh water flux terms, we have used high resolution winds ($0.312°\times0.312°$ spatial grid) from CFSR (Climate Forecast System Reanalysis).

**SC#3:**

L108-115 – The AVISO SLA is HIGHLY suspect in the Red Sea for resolving eddies. Yes, they grid it at quarter degree spacing, but how much actual data is there? Also, their covariance functions for the OI are not tuned to the Red Sea in general, or to the Tokar Gap in particular. See below, too, but I would remove the AVISO SLA eddy material and focus on the Tokar Gap analysis that doesn't require it.

Reply:

The SLA data from AVISO is used just for a broad and qualitative understanding on the changes in sea level in the Red Sea. We have used the merged data from all satellite estimates. Red Sea has considerable number of satellite tracks from different satellites (Fig. ). Further, the AVISO SLA data have been used by previous studies also for the Red Sea region, for example Zhan et al., (2014), Papadopoulos et al., (2015) and Taqi et al., (2017).

[Figure]

Fig.1 Satellite tracks in the Red Sea.

We agree that a more precise and quantitative estimation of the sea level variability may require further improvements in the AVISO product. But, the SLA data are providing a qualitative understanding of the sea level changes in the Red Sea. We have also compared the geostrophic currents from the hydrographic measurements (profiles collected during different cruises) and from AVISO SLA. Both are matching well. Apart from this, the ADCP measurements carried out by Sofianos and Johns (2007) show the presence of eddies in the

Red Sea, which are well matching with gridded SLA estimates from AVISO for the same period. This gives us confidence to use SLA estimates in the present analysis, at least for a qualitative understanding of the sea level variability.

**SC#4:**

L122-129 – Well-done to switch to an along-axis coordinate system and to do the analysis in an along-axis, time space. Nice!

Reply:

Thank you for appreciating the work.

**SC#5:**

L150-162 – I'm worried that we are over-interpreting in this section. The general seasonal pattern in clear, but the along-axis changes are not so clear. Without error bars it is difficult to tell when we are interpreting real changes, or just noise.

Reply:

The error (observed standard deviation from monthly mean value) is mentioned in the text at appropriate locations in the revised manuscript. For the entire climatology, the standard deviation is less than 10 m for >95% cases, while active mixing zone show relatively higher standard deviation.

The observed relatively larger deviation during winter especially in the northern latitudes is due to the measurement of profiles during the ongoing mixing process. In addition, the convection process in the northern Red Sea show considerable interannual variability. This resulted in wide range of MLD values and relatively large standard deviation from the mean value.

The modified text in the manuscript:

A Hovmoller diagram of the monthly MLD climatology is presented in Fig. 3. The deepest MLD is observed in February and the shallowest during May-Jun. A significant annual variability is observed in the Red Sea. The maximum value of climatological mean MLD is observed in February at the northern Red Sea while the minimum noticed at various instances, especially during summer months. The MLD of individual profiles in the northern Red Sea has a wide range values from 40 to 120 m mainly due to the presence of active convection process, while some of the profiles show MLD deeper than 150 m in consistence with Yao et al., (2014). Apart from the northern deep convection region, the south-central Red Sea between 18 °N-21 °N (53±5 m) and 14 °N-16 °N (48±9 m) also experienced deeper MLDs during the winter, which is separated by a shallower MLD around 17 °N (44±14 m). During July to September, the region around 19° N experienced a deeper mixed layer in contrast with the general pattern of summer shoaling over the entire Red Sea.

The deepening of the MLD begins in October throughout the Red Sea. The winter cooling and its associated convection strengthen by December, with an average MLD>50 m. Compared to other parts 201 of the Red Sea, during November and December, relatively shallower MLDs were witnessed at approximately 16° N-17° N, and 24.5° N-26.5° N. The winter deepening of the MLDs intensifies by January and continues throughout February. In contrast to the general pattern of deeper MLDs in the 204 northern latitudes, the area between 24.5° N and 26.5° N shows a relatively shallow MLD almost throughout the year, especially in the winter.

The mixed layer starts to shoal gradually by the end of February, and the MLDs of most areas decreases to 20±7 m by April. Summer shoaling is comparatively stronger in the 15° N-18° N latitude band, and the detected mean MLD is < 15 m. Individual observations revealed that many profiles have MLDs < 5 m. In general, the shallow mixed layers are predominant from April to September, while this prevails until October in the far north. In the south-central Red Sea, the shallow mixed layer exists for only a short period, from April to June.

**SC#6:**

L160 – As an example of the comment I just made, I cannot make any sense of a "general pattern of deeper MLDs in the latitudes". I just cannot see in Figure 2 where this statement comes from! You will need to be much more specific to convince me of this and it will also require appropriate error estimates. You lose little by sticking to the bigger picture and skipping the "wiggles", although the "tongue" at the latitude of the Tokar Gap is interesting (see below).

Reply:

The latitudinal variability in the MLD is clear during winter, with deepest MLD in the north and shallow in the south with deeps/shallows in between. Due to this reason, we used the term "general pattern of deeper MLDs in the northern….". We have removed this part of the sentence from the text.

The earlier text in the manuscript:

In contrast to the general pattern of deeper MLDs in the northern latitudes, the area between 24.5° N and 26.5° N shows a relatively shallow MLD almost throughout the year, especially in the winter.

The modified text in the manuscript:

The area between 24°N and 27°N shows a relatively shallow MLD almost throughout the year, especially during winter.

**SC#7:**

L205 and Figure 5 – I like this plot, but some estimate of the degrees of freedom needs to be made so that we know whether the structure in the curve real or just statistical fluctuations. And, as usual, the degrees of freedom estimate should consider red noise and not assume white, or independent, noise.

Reply:

We have tested the statistical significance of the correlation values. The estimated p-value, t-value and the effective degree of freedom show that the correlation values are significant at 95%. We have tabulated the above stated parameters for a single case (for correlation between NHF and MLD) in the Table given below.

Table.1 Statistics for the correlation between NHF (net heat flux) and MLD.

| Latitude (N) | P-value | Effective degree of freedom (timesteps=420. 35*12 months) | t-value based on Bretherton et al, (1999) | t-value for 95% confidence level from "T table" |
|---|---|---|---|---|
| 13 | 3.20E-09 | 269.6597 | 4.842266 | 1.650517 |
| 13.5 | 0.002644 | 282.5494 | 2.477852 | 1.650256 |
| 14 | 1.38E-06 | 243.9125 | 3.726658 | 1.651123 |
| 14.5 | 5.33E-19 | 237.3118 | 7.015846 | 1.651308 |
| 15 | 3.80E-20 | 218.1968 | 6.965186 | 1.651873 |
| 15.5 | 1.76E-12 | 174.2134 | 4.667594 | 1.653658 |
| 16 | 6.64E-23 | 219.8966 | 7.552761 | 1.651809 |
| 16.5 | 2.61E-32 | 222.1904 | 9.367547 | 1.651746 |
| 17 | 2.53E-56 | 189.838 | 12.41162 | 1.652913 |
| 17.5 | 7.79E-41 | 182.2344 | 9.824286 | 1.653269 |
| 18 | 6.43E-80 | 163.4437 | 14.81243 | 1.654256 |
| 18.5 | 8.99E-47 | 183.8425 | 10.77849 | 1.653177 |
| 19 | 6.06E-45 | 178.4458 | 10.34322 | 1.653459 |
| 19.5 | 2.85E-72 | 164.7387 | 13.79153 | 1.654141 |
| 20 | 5.32E-85 | 159.815 | 15.35839 | 1.654433 |
| 20.5 | 2.37E-86 | 156.5116 | 15.38553 | 1.654617 |
| 21 | 2.27E-74 | 203.4905 | 15.67503 | 1.652394 |
| 21.5 | 3.67E-92 | 156.3192 | 16.19286 | 1.65468 |
| 22 | 7.40E-92 | 144.0271 | 15.4933 | 1.655504 |
| 22.5 | 4.43E-56 | 204.1266 | 12.83679 | 1.652357 |
| 23 | 4.79E-56 | 237.2501 | 13.84296 | 1.651308 |
| 23.5 | 1.15E-65 | 302.7621 | 17.48819 | 1.649898 |
| 24 | 5.63E-112 | 139.1993 | 17.98179 | 1.65589 |
| 24.5 | 1.68E-87 | 144.9487 | 14.95348 | 1.65543 |

| | | | |
|---|---|---|---|
| **25** | 4.46E-87 | 216.5994 | 18.25168 | 1.651906 |
| **25.5** | 1.10E-53 | 218.3447 | 12.89109 | 1.651873 |
| **26** | 2.70E-51 | 128.179 | 9.546686 | 1.656845 |
| **26.5** | 6.08E-35 | 179.0278 | 8.820048 | 1.653411 |
| **27** | 1.74E-74 | 122.4401 | 12.13284 | 1.657439 |
| **27.5** | 1.35E-78 | 151.0859 | 14.05475 | 1.655007 |

**SC#8:**

L211-215 – I think we are over-interpreting again. These are very small "wiggles" and 2 of the 5 do not even show the pattern you assert, meaning all three curves moving down together. Again, if you want to interpret these small changes, then you have to do a much more thorough job on the statistics to convince us that we're not just looking at random fluctuations.

Reply:

As mentioned in reply to the previous comment (SC#7), the observed fluctuations are statistically significant at 95% confidence level. The statistical results based on p-value, t-test and degrees of freedom has shown that the parameters (heat flux, freshwater flux and wind stress) correlation coefficients are significant at 95% confidence level.

We have repeated the analysis after smoothening MLD climatology for 1 degree along the latitude and figures are given below.

[Figure]

Fig.2 MLD climatology smoothed along latitude for 1 degree.

[Figure]

Fig.3 a) The correlation between MLD and atmospheric forces for smoothed MLD climatology, and b) the number of eddies in the Red Sea for the period 1992-2012.

The MLD climatology and correlation curves show a smoothed, but similar structure. A decreasing pattern can be seen correlation values at 19N (clear in wind-stress and heat flux), at 23N (clear in all three forces) and at 26.5N (clear in heat flux and freshwater flux). The correlation drops around 13.5N and 17.5N are less visible. Additionally, a drop around 15N also can be seen associated

Believing that the smoothening may remove some of the small-scale features, we would like to use the original MLD climatology (without smoothening) and previous version of the correlation curve in the manuscript.

**SC#9:**

L216, all of Section 3.3 – As I mentioned above, the analysis using the AVISO SLA is highly questionable here. I think this entire section, and basically all use of AVISO, is not necessary for this paper. You have some good results, but by pushing too far you run the risk of most readers doubting everything. Please note that I am trying to be constructive here and help to improve the paper. I like the paper as a whole, but really do not like this section.

> Reply:
>
> The Sea level anomaly estimate from AVISO is a merged product of multiple satellite tracks. It is true that the research based on SLA has to be carried out with caution, especially in smaller regions like Red Sea. As mentioned in reply to comment SC#3, the Red Sea has considerable number of satellite tracks. Further, multiple studies have already been carried out based on this data. The previous studies show that AVISO SLA estimates can still provide the general picture of sea level changes in the Red Sea. Our study, based on SLA, is only looking to the main locations of eddies in the Red Sea.

**SC#10:**

L258-260, Figure 6 – This is a continuation of the previous comment. You say that there is a good match with the number of eddies and the latitude ranges you identified earlier from the correlation curves, but I simply do not see that. And since we're just doing the analysis "by eye", then my eye is as good as yours. You really have to do some statistics if you want to make this point. And once again, none of these things are core results of your paper.

> Reply:
>
> We agree that few of the correlation drops are not matching with the eddy locations. We have mentioned the same in the manuscript also.

The correlation drops locations matching with eddies:

1. At 19°N: matching with the locations of Tokar region and eddies observed by Zhan et al 2014.
2. At 23°N: matching with eddies observed by Zhan et al 2014.
3. At 26.5°N: matching with eddies observed by Papadopoulos et al., 2015

The correlation drops locations not matching with eddies:

1. At 13.5°N: The Red Sea is very narrow at 13.5°N and close to Bab-el-Mandab strait. Moreover, complex dynamics associated with the exchange of surface and subsurface waters between the Red Sea and the Gulf of Aden occurs in this region. The complexity of this region prevents linking the MLD variability directly to atmospheric forcing or eddies.
2. At 17.5°N: The region at approximately 17.5° N is between the two eddy-driven downwelling zones at approximately 15° N and 19° N (Fig. 2). Mass conservation requires upwelling to replace the downwelling water. The MLD climatology shows shallow mixed layers throughout the year at 17.5° N, which could be due to possible upwelling. Further investigation is required to unveil the dynamics associated with this region.

The drops in correlation at 23°N and 26.5°N are matching with eddy locations. The drop at 19°N is matching with the Tokar region and eddies. indicating the effect of eddies. As mentioned above, the other two locations (13.5°N and 17.5°N) need further investigations to unveil the associated dynamics.

**SC#11:**

L277, all of Section 3.4 – Much better! Looking at the 19N signal where the Tokar Gap is, and showing the actual wind results rather than AVISO eddy counts is much more convincing. And note that this is the only latitude band where I see convincing results. It's well-known that strong winds through mountain gaps generate the eddy signals you infer (search for results on the Hawaiian Ridge and the Gulf of Tehuantepec), so you do not need the questionable AVISO results in order to rationalize the "tongue" in Figure 2 at 19N. This is a very nice result.

Reply:

Thank you very much for your appreciation.

We agree that mountain gap winds can generate eddy signals in the underlying sea. Therefore, the SLA snapshots are not necessary to show the impact of Tokar winds and associated deepening in MLD to the right of the wind jet and shoaling to the left. We keep SLA maps in figure just for helping the reader to easily understand the position of profiles influenced by mountain gap winds.

**References:**

1. Al Saafani, M. A., & Shenoi, S. S. C.: Water masses in the Gulf of Aden. *Journal of oceanography*, *63*(1), 1-14, 2007.
2. Bretherton, C. S., Widmann, M., Dymnikov, V. P., Wallace, J.M. & Blade, I. The effective number of spatial degrees of freedom of a time-varying field, J. Clim., 12, 1990–2009 (1999).
3. Papadopoulos, V. P., Zhan, P., Sofianos, S. S., Raitsos, D. E., Qurban, M., Abualnaja, Y., Bower, A. S., Kontoyiannis, H., Pavlidou, A., Asharaf, T. T. M., Zarokanellos, N. and Hoteit, I.: Factors governing the deep ventilation of the Red Sea, J. Geophys. Res. Ocean., 120(11), 7493–7505
4. Sofianos, S. S. and Johns, W. E.: Observations of the summer Red Sea circulation, J. Geophys. Res. 450 Ocean., 112(6), 1–20, doi:10.1029/2006JC003886, 2007
5. Taqi, A. M., A. M. Al-Subhi & M. A. Alsaafani (2017): Extension of Satellite Altimetry Jason-2 Sea Level Anomalies Towards the Red Sea Coast Using Polynomial Harmonic Techniques, Marine Geodesy, DOI: 10.1080/01490419.2017.1333549
6. Zhan, P., Subramanian, A. C., Yao, F. and Hoteit, I.: Eddies in the Red Sea: A statistical and dynamical study, J. Geophys. Res. Ocean., 119(6), 3909–3925, doi:10.1002/2013JC009563, 2014.

---

## Author Response (AR2)

**Response to the Topic Editor Comments**

**Contents:**

**1- Response to the review comments**

**Decision**: Publish subject to minor revisions (review by editor) (04 Jun 2018) by Piers Chapman

Thank you very much for your valuable comments and suggestions on this manuscript, entitled "Mixed layer depth variability in the Red Sea". The comments and suggestions were very helpful in improving the manuscript. The manuscript is modified according to the comments and the changes are given below.

Please note that the manuscript with tracked changes is given in this document itself, after the list of response to the comments.

**Comment#1**

I think that if you want to use the AVISO data, from which most of the information of the eddy field is derived, then you need to at least point out the deficiencies as detailed by reviewer #2 in section 2.2. I am also not very enthusiastic about some of the description in section 3.3; I don't see any sign in Fig. 7, for example, of eddies that are supposed to exist near 13°, 17° or 26°N, even though this has been reported by others, but there does seem to be a small increase near 15°N in this figure. So I think this section could be shortened and made less important.

**Answer:**

**Part#A:** The manuscript is modified accordingly. The number of satellite tracks are relatively lower in the narrow regions like Red Sea and we have mentioned the same in the manuscript also. Even though, the merged satellite product is helpful for a qualitative understanding on the sea level variability in the Red Sea.

*[Line number in the clean manuscript: 108-110]*

*[Line number in the manuscript with tracked changes: 130-132]*

**Part#B:** Agreeing to the Editor comment, the section 3.3 is removed and a shortened form of this section. The description of the impact of eddies in more than 60 lines (in the previous version of the manuscript, lines 235-295) is shortened to just 18 lines (in the revised manuscript, lines 238-255) and merged to section 3.2.

*[Line number in the clean manuscript: 238-255]*

*[Line number in the manuscript with tracked changes: 305-322]*

**Comment#2**

You discuss the MLD climatology in Fig. 3, and stress the importance of the winter minima near 17°N and 25°N. However, from Fig. S3, January showed the fewest samples, so are these minima really significant given that you have 29 latitudinal bands of 0.5° each?

**Answer:**

The noise in mean MLD for the region around 25N is relatively small (~30+/-9 m) comparing to the difference in MLD values towards northern (~70m) and southern (~50m) grids. So this shallow MLD can be considered.

At 17N, the noise in MLD (44+/-14m) is overlapping with mean MLD of northern (~53m) and southern (~48m) grids. Therefore, additional in-situ data is required to confirm (which may reduce the noise) the observed shallow in this region. So this shallow MLD can be excluded from the manuscript.

The manuscript modified accordingly, to keep the shallow MLD around 25N and exclude the "shallow MLD around 17N".

*[Line number in the clean manuscript: 164-168]*

*[Line number in the manuscript with tracked changes: 206-210]*

**Comment#3**

I also had some problems with section 3.2, particularly the relationships between the MLD and the forcing functions shown in Fig. 6. In your response to reviewer #2, you said that you have tested these relationships statistically and that they are all significant, yet you don't say this in the paper. So say so, otherwise they are just wiggles in the data.

**Answer:**

The statistical significance of the correlation values are verified based on T-test following Bretherton et al, (1999)), and the estimated p-value, t-value and the effective degree of freedom show that the correlation values are statistically significant at 95%.

The manuscript modified accordingly.

*[Line number in the clean manuscript: 223-226]*

*[Line number in the manuscript with tracked changes: 289-292]*

**Comment#4**

Finally, in section 3.4, you should reference some of the work in similar areas such as the Gulf of Tehuantepec, where strong winds coming through mountain passes are known to affect mixing.

**Answer:**

We have included appropriate reference in the manuscript.

*[Line number in the clean manuscript: 313-316]*

*[Line number in the manuscript with tracked changes: 522-525]*

**Comment#5**

Lines 34-37 – suggest you rewrite as: "The Red Sea is an important intermediate water formation region in the world ocean. Red Sea Outflow Water (RSOW), formed mainly due to open-ocean convection in the northern Red Sea (Sofianos and Johns, 2002), propagates through Bab-el-Mandab to the Gulf of Aden (A&S 2007) and later spreads to the Indian Ocean. Its signature reaches….."

**Answer:**

The manuscript modified accordingly.

*[Line number in the clean manuscript: 34-38]*

*[Line number in the manuscript with tracked changes: 35-39]*

**Comment#6**

Lines 49-50 – suggest "The Red Sea has been investigated for many years with an emphasis on its different physical features, but there has been no detailed investigation on MLD variability, apart from a few studies addressing the hydrography…."

**Answer:**

The manuscript modified accordingly.

*[Line number in the clean manuscript: 49-52]*

*[Line number in the manuscript with tracked changes: 58-61]*

**Comment#7**

Line 64: End the sentence after "is the main source." and delete "with larger number of profiles."

**Answer:**

The manuscript modified accordingly.

*[Line number in the clean manuscript: 64]*

*[Line number in the manuscript with tracked changes: 73]*

**Comment#8**

Lines 125-127 – delete the last sentence of this paragraph ("This method first identifies….") as it merely repeats what you have said already.

**Answer:**

The manuscript modified accordingly.

*[Line number in the clean manuscript: 128]*

*[Line number in the manuscript with tracked changes: 158]*

**Comment#9**

Lines 162-163 – suggest you talk about the region between 14°-21°N as a whole, rather than splitting it up (see my comment about MLD climatology above). Is a change from 48+/- 9 to 44 +/- 14 really significant? Should the reference in line 162 be to Yao et al 2014b?

    **Answer:**

    *Part#A: The manuscript modified accordingly. As suggested in the comment, the noise in mean M LD around 17°N is significantly high. Therefore, the text is corrected accordingly and this shallow region is not considered.*

    *[Line number in the clean manuscript: 164-168]*

    *[Line number in the manuscript with tracked changes: 206-210]*

    **Part#B:** The reference is corrected as Yao et al 2014b.

    *[Line number in the clean manuscript: 163]*

    *[Line number in the manuscript with tracked changes: 205]*

**2- The manuscript with tracked changes**

[revised manuscript text omitted]